# VideoSearch Reasoner: Boosting Multimodal Reward Models through Thinking-with-Image Reasoning

## Abstract

Recent advancements in multimodal reward models (RMs) have substantially improved post-training for visual generative models. However, current RMs face inherent limitations: **(1)** visual inputs consume large context budgets, forcing fewer frames and causing a loss of fine-grained details; and **(2)** all visual information is packed into the initial prompt, exacerbating hallucination and forgetting during chain-of-thought reasoning. To overcome these issues, we introduce VideoSearch Reasoner, a thinking-with-image framework that equips the RM with visual reasoning operations (e.g., select frame) and a configurable visual memory window. This allows the RM to actively acquire and update visual evidence within context limits, improving reasoning fidelity and reliability. We activate visual reasoning via a reinforcement fine-tuning pipeline: **(i)** Cold Start with curated visual chain-of-thought data to distill basic reasoning skills and operation formatting; **(ii)** select samples whose per-dimension and overall judgments are all correct, then conduct Rejection sampling Fine-Tuning on these high-quality traces to further enhance reasoning; and **(iii)** apply Group Relative Policy Optimization (GRPO) to strengthen reasoning. Our approach delivers state-of-the-art accuracy among open-source models on video preference benchmarks, especially for longer videos: a 7B VideoSearch Reasoner achieves 80.5% on VideoGen Reward, 82.3% on GenAI-Bench, and 75.6% on MJ-Bench-Video. These results validate the effectiveness and promise of thinking-with-image multimodal reward modeling.

## 1 Introduction

With the advancement of multimodal Reward Models (RMs) (Wang et al., 2025b; Zang et al., 2025; Wang et al., 2024; Xiong et al., 2024; Liu et al., 2025b; Xu et al., 2024; He et al., 2024), the substantial potential of RMs in aligning vision models with human preferences (Liu et al., 2025a;b; Schulman et al., 2017; Ouyang et al., 2022) has garnered increasing attention, owing to their capacity to provide accurate reward signals during model training and fine-tuning processes (Liu et al., 2024; Wijaya et al., 2024). Most RMs are predominantly classifier-based or generative (Xiong et al., 2024; Wang et al., 2024; Li et al., 2025; Liu et al., 2025b; Wang et al., 2025c; Tong et al., 2025; Zang et al., 2025). After being trained on large, pre-annotated preference datasets, they typically either (i) directly output scalar scores (and, for pairwise data, relative preference rankings), or (ii) produce brief natural-language justifications along with judgments. The former mode tends to operate as a black box, raising concerns about insufficient interpretability; the latter often relies on rudimentary reasoning, lacking concise logical structure and depth of analysis, thereby undermining accuracy.

In light of these issues, recent works (Wu et al., 2025a; Wang et al., 2025b; Hong et al., 2025; Chen et al., 2025) have proposed reasoning-based RMs to leverage the language generation capabilities of Visual Language Models (VLMs). By eliciting richer chains of reasoning, these approaches aim to produce multi-dimensional, logically structured, and more in-depth analyses, thereby improving the accuracy, robustness, and transparency of RMs. Despite these successes, inherent limitations remain in VLM-based RMs, particularly for video preference data. On the one hand, visual inputs consume substantial context budget, forcing RMs to process fewer frames and risking the loss of fine-grained details. On the other hand, all visual information is typically packed into the initial prompt; during

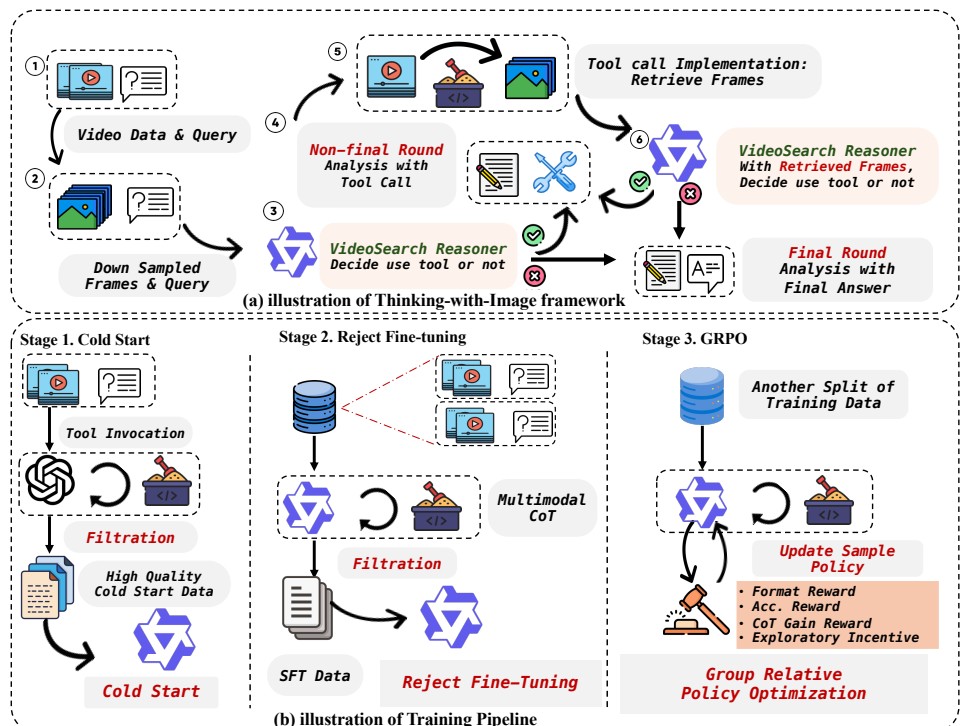

**Figure 1:** (a) shows the main process of our proposed Thinking-with-Image framework. (b) shows an overview of the three training stages we proposed, including Cold Start, Rejection sampling Fine-Tuning, and GRPO.

the RM's Chain-of-Thought (CoT) reasoning, the process proceeds purely in text without revisiting or updating visual evidence, which exacerbates forgetting and hallucination.

In this work, we introduce a novel **thinking-with-image** framework to address the aforementioned concerns by equipping the RM with visual reasoning operations like frame selection and a configurable visual memory window (Wang et al., 2025b; Guo et al., 2025a; Su et al., 2025a). Frame selection enables the model to actively retrieve previously seen frames and acquire unseen visual evidence as new inputs to subsequent reasoning rounds, thereby improving fidelity. The configurable memory window retains only the most recently active visual information, ensuring that, under context-length constraints, the model can select frames multiple times, broaden its visual field, and extend both its reasoning horizon and the total number of frames it can process, while keeping the memory footprint stable. Building on this framework, we propose VIDEOSEARCH REASONER, the first multimodal RM capable of visual reasoning. In principle, it imposes no upper bound on the number of frames it can process, enabling fidelity-preserving evaluation for long video reward tasks.

Specifically, the training pipeline comprises three stages: **(I) Cold Start.** Using curated visual CoT data, we instill basic textual reasoning skills and operation formatting (e.g., invoke frame selection). **(II) Rejection sampling Fine-Tuning.** We run the model on large-scale preference datasets, which include fine-grained, per-dimension assessments alongside an overall judgment. We then retain only samples with all judgments correct, and conduct Rejection sampling Fine-Tuning on these verified traces to encourage accurate, high-quality visual and textual reasoning. **(III) Group Relative Policy Optimization (GRPO).** We apply GRPO on collected preference data, incentivizing the model to explore details in videos and optimize toward reward rules that favor high-quality reasoning with correct per-dimension and overall judgments. In summary, our contributions are as follows:

- We propose VideoSearch Reasoner, the first multimodal RM capable of visual reasoning, which substantially alleviates context-length constraints and mitigates forgetting of visual information.

- In VideoSearch Reasoner, we propose to equip the RM with visual reasoning operations like frame selection and a configurable visual memory window based on thinking-with-image framework.

- We demonstrate the crucial role of visual reasoning in multimodal RMs, showing improved accuracy and reliability on preference datasets and significantly increased usability and fidelity.

## 2 RELATED WORK

**Multimodal Reward Models** (RMs) have garnered increasing attention (He et al., 2024; Liu et al., 2025b; Xu et al., 2024; Wang et al., 2025b) for their potential to effectively optimize vision generation models to better align with human preferences. Vision-language models (VLMs) (Bai et al., 2025; Bordes et al., 2024), have become the models of choice for RMs. For instance, Liu et al. (2025b) proposes VideoReward, a reward model that directly regresses preference-aligned scores from input videos; Wang et al. (2025c) develops UnifiedReward in a generative response format. However, such approaches often lack rigorous logical structure and deep analysis. To this end, Wang et al. (2025b) introduces a reasoning framework in multimodal RMs, aiming to improve the accuracy of reward signals. Despite these advances, VLM-based RMs still face inherent limitations, especially on video preference datasets with more frames and longer durations (Liu et al., 2025b; Tong et al., 2025). Specifically, first, visual inputs consume substantial context budgets, forcing RMs to subsample only a subset of frames and thereby losing fine-grained details (Tong et al., 2025; Liu et al., 2025b; Wang et al., 2024; He et al., 2024; Xu et al., 2024). Second, during the RM's generative response, reasoning proceeds purely in text without revisiting or updating visual evidence (Wang et al., 2025b;c).

**Thinking-with-Image** is an emerging paradigm in VLM reasoning that overcomes the limitations of text-centric chains of thought that treat visual inputs merely as a static initial context (Shen et al., 2024; Mallis et al., 2024; Xu et al., 2025; Duan et al., 2025; Su et al., 2025b). Instead, it treats vision as a dynamic, operable cognitive workspace, leveraging visual information throughout intermediate reasoning steps. Two primary modes characterize this paradigm: **(1) Intrinsic imagination**, which allows the model to reason directly over the corresponding visual tokens (Team, 2024; Xu et al., 2025; Guo et al., 2025b). **(2) Active exploration**, which enables the model to proactively retrieve visual information via *toolchain invocation* (the VLM calls external tools through a specified interface) or *programmatic manipulation* (the VLM emits executable code that directly defines operations) (Shen et al., 2024; Mallis et al., 2024; Wang et al., 2025a;d).

**Video Retrieval-Augmented Generation** (Video RAG) extends the traditional RAG paradigm to video understanding by coupling a retrieval module with a visual language backbone to answer queries across extended temporal contexts (Ren et al., 2025; Jeong et al., 2025; Luo et al., 2024; Sagare et al., 2024; Gao et al., 2024). These systems index videos into semantically coherent segments and retrieve those most relevant to an *external textual query* (Luo et al., 2024; Ren et al., 2025; Sagare et al., 2024). Such architectures exemplify *passive, query-driven systems*: the retrieval process is initiated by the user's prompt, and relevance is determined through predefined similarity metrics. In contrast, the **Thinking-with-Image** framework departs from this retrieval-centric design by enabling **active, reasoning-driven exploration** of visual information. Instead of merely fetching visual evidence for an externally defined query, the model autonomously hypothesizes, inspects, and refines its visual focus throughout the reasoning process.

## 3 VIDEOSEARCH REASONER

In this section, we first elaborate on the concrete components of the Thinking-with-Image framework (Section 3.1). We then present the multi-stage training pipeline, explaining how it elicits and cultivates multimodal reasoning capabilities in both vision and text (Section 3.2).

### 3.1 THINKING-WITH-IMAGE-BASED FRAMEWORK

The data flow of VIDEOSEARCH REASONER under our Thinking-with-Image framework is shown in Figure 1. Video preference data are uniformly downsampled into a preset number of input frames as visual input and paired with a prompt template that explicitly specifies the total number of frames and the downsampling scheme. The model then iteratively performs tool invocations and updates its reasoning with the tool-execution outcomes; these outcomes remain valid only within a preset window. To mitigate the risk of information loss, the reasoning format converts visual evidence into linguistic summaries within specific regions.

**Tool Invocation.** Consistent with standard VLMs used as reward models, our model requires downsampling for video inputs. However, instead of treating the unselected visual content as if it does not exist, we retain it as an operable workspace that the model is aware of. After an initial round of

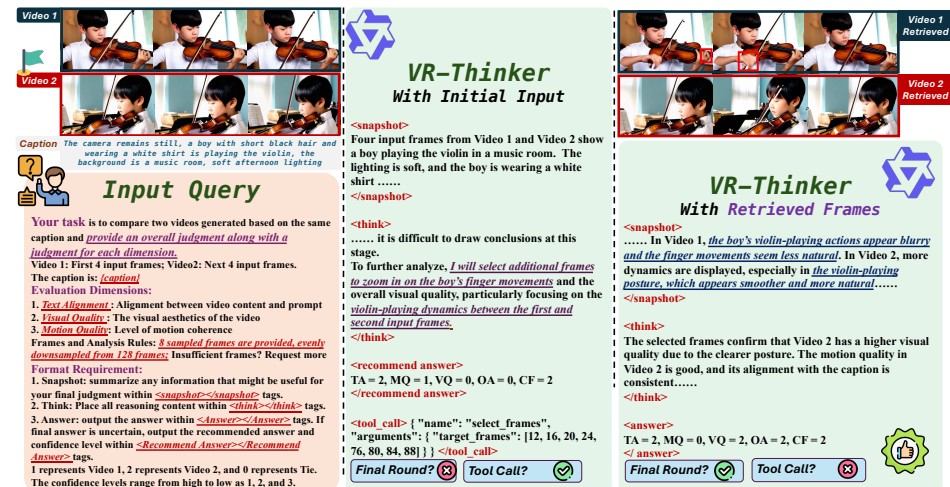

**Figure 2:** Qualitative Cases. When frames are down sampled, key information might not be included in the input. VIDEOSEARCH REASONER actively retrieves frames, which ensures the correctness of such cases.

multimodal reasoning, the model may find that missing evidence (or near-ties in paired preference cases) prevents a definitive judgment, which could potentially be due to the unselected visual information. The model then iteratively issues tool-invocation instructions to retrieve additional visual evidence, and updates its reasoning by incorporating the tool-execution outcomes, repeating this process until a final preference judgment can be made.

Formally, the initial input is $\mathcal{X} = [\mathcal{V}, T]$, where $\mathcal{V}$ is the downsampled visual input and $T$ is the textual query. A model $\pi_\theta$ constructs a multimodal reasoning chain via iterative reasoning and tool invocation, where at each step the model first yields a reasoning unit $r_t \sim \pi_\theta(\cdot \mid \mathcal{X}, \tilde{R}_{t-1})$, conditioned on the initial input $\mathcal{X}$ and all preceding $t-1$ reasoning steps denoted by $\tilde{R}_{t-1}$. Each reasoning unit may be purely textual or multimodal; the latter can then invoke a tool to directly interact with information in the whole visual workspace ( denoted by $\tilde{\mathcal{V}}$, to distinguish $\mathcal{V}$). For each multimodal reasoning step $r_t$, the model calls a tool $f$, obtains a tool-execution outcome $o_t = f(\tilde{V})$, for subsequent reasoning steps.

**Window Memory**. The reasoning process does not, by default, retain all tool-execution outcomes. Instead, we employ a windowed memory: each outcome remains active for a preset number of rounds $p$ before being deliberately forgotten. This design is motivated by the substantial context budget consumed by visual information, especially for longer videos where frames dominate the context: In multimodal reasoning, the textual portion per segment $R_n$ typically occupies less than 400 tokens, while a single visual frame contributes roughly 500 tokens. With a default of 8 input frames, visual evidence accounts for approximately 4,000 tokens, around $10\times$ the textual budget. Under the windowed memory, the total context usage remains relatively stable, preventing bottlenecks from repeatedly retrieving additional visual information through tool invocation.

Formally, after each update, we maintain the entire prefix of reasoning units but only with a sliding window over the most recent tool outcomes: Let $\tilde{R}_{t-1}$ denote the prior reasoning chain, $r_t$ the new reasoning unit. The update process can be described as:

$$\tilde{R}_{t-1} = [r_1, r_2 \ldots, r_{t-p-2}, (r_{t-p-1}, o_{t-p-1}), \ldots, (r_{t-1}, o_{t-1})]$$

$$r_t \sim \pi_\theta(\cdot \mid \mathcal{X}, \tilde{R}_{t-1}), \quad \text{where tool } f \text{ is called}$$

$$o_t = f(\mathcal{V})$$

$$\tilde{R}_t = [r_1, r_2 \ldots, r_{t-p-1}, (r_{t-p}, o_{t-p}), \ldots, (r_t, o_t)]$$

, where $p$ is the window width and $(r_k, o_k)$ denotes a reasoning unit paired with its tool-execution outcome retained within the window. The total token count $\mathcal{T}_{\text{total}}$ till step $t$ is

$$\mathcal{T}(\mathcal{X}) + \mathcal{T}(\tilde{R}_t) = \mathcal{T}(\mathcal{V}) + \mathcal{T}(T) + \sum_{k=1}^{t} \mathcal{T}(r_k) + \sum_{k=t-p}^{t} \mathcal{T}(o_k) \approx \mathcal{T}(\mathcal{V}) + \sum_{k=t-p}^{t} \mathcal{T}(o_k),$$

where $\mathcal{T}(\cdot)$ denotes the number of tokens and we approximate textual tokens as a minor component relative to visual tokens. Further, approximating token costs by per-frame contributions, we obtain $\mathcal{T}_{\text{total}} \approx (N_{\text{in}} + pN_{\text{ex}})V_t$, where $N_{\text{in}}$ is the number of initial input frames, $N_{\text{ex}}$ is the number of frames retrieved per tool invocation, $p$ is the window width, and $V_t$ is the average token cost per visual frame. Crucially, $\mathcal{T}_{\text{total}}$ is approximately independent of the total number of reasoning steps $t$, highlighting how windowed memory sustains the context budget under this setting.

**Reasoning Format.** As shown in Figure 2, the model is required to follow a specific reasoning format, using XML-style tags to *delineate* functional areas and reasoning-focus categories, which helps ensure clarity and consistency in reasoning and logical structure.

In addition to commonly used tags like `<think>` and `<answer>` in reasoning models, two additional tags are employed: **`<Snapshot>`**: This tag is used in every reasoning segment to mitigate the risk of forgetting critical information under the *Window Memory* mechanism. After each execution outcome is incorporated, this tag is used to create a snapshot of essential information from these frames in the form of language tokens. This approach serves as an information compression strategy, reducing thousands of visual tokens to dozens of language tokens, which balances *fidelity* and *budget*. **`<Recommend Answer>`**: Unlike the `<answer>` tag, this tag is used in non-final reasoning segments. The model outputs its current preferred result along with the confidence level, which helps assess the value of additional multimodal reasoning segments and also aids the model in organizing its current judgments.

## 3.2 MULTI-STAGE REWARD MODEL TRAINING

The training pipeline consists of three main stages: (i) *Cold Start* efficiently elicits textual reasoning skills and bootstraps basic visual reasoning. (ii) *Rejection sampling Fine-Tuning* consolidates both textual and visual reasoning capabilities. (iii) *Exploratory Reinforcement Learning* reinforces the integrated multimodal reasoning ability.

### 3.2.1 COLD START & REJECTION FINE-TUNING

**Cold Start.** This stage serves two purposes. First, VLMs have limited zero-shot ability to execute novel tool invocations. To ensure accurate reasoning structure and tool-calling syntax, we employ CoT data that adheres to our reasoning format. Second, although VLMs possess strong latent linguistic reasoning capabilities, inadequate reward modeling often leads to underdeveloped reasoning behavior. High-quality Cold Start CoT data not only elicits linguistic reasoning but also bootstraps basic visual reasoning through vision-related analytical steps embedded in the trajectories.

Concretely, we construct Cold Start data by selecting a subset of video pairs and textual queries from a video preference dataset. Following the Think-with-Image framework, we iteratively invoke a powerful multimodal model, e.g. GPT-4o (Hurst et al., 2024), to generate high-quality, long CoT trajectories. A two-stage filtering process ensures that these multimodal CoTs are suitable for initialization: (i) every reasoning segment must strictly conform to the prescribed format, and (ii) the final judgments, both per-dimension and overall preference, must exactly match the ground-truth labels in the preference dataset, thereby guaranteeing high-accuracy multimodal reasoning. We train with the standard Supervised Fine-Tuning (SFT) loss during this Cold Start phase, while masking tokens associated with tool-execution outcomes from the loss computation.

**Rejection Sampling Fine-Tuning.** The previous stage instilled the correct reasoning format and high-quality multimodal CoT exemplars, initializing the model's reasoning capabilities. However, the proportion of model-generated CoT samples that are both well-formed and accurate remains low. An excess of negative samples due to limited Cold Start data and training epochs hampers the efficiency of sampling-based reinforcement learning. To consolidate the learned reasoning skills and increase the yield of high-quality reasoning segments, thereby paving the way for RL. We perform Supervised Fine-Tuning on a large, rejection-sampled multimodal CoT dataset.

Specifically, we blend multiple video preference datasets and select a large subset of video–query pairs. Similar to the previous stage, we generate CoT samples, but now we sample from the model trained in Stage 1, drawing multiple samples per input to ensure sufficient positives. The same two-stage filtering is applied to construct the SFT dataset. We use the same loss as in the Cold Start

phase, with tool-execution outcome tokens masked from the loss. This stage substantially improves both the format compliance and quality of the model's reasoning segments.

### 3.2.2 EXPLORATORY REINFORCEMENT FINE-TUNING

To further reinforce multimodal reasoning on top of these capabilities we apply GRPO-based reinforcement fine-tuning. Using predefined rule-based reward functions together with additional exploratory incentives, we evaluate the model-sampled reasoning segments and iteratively optimize the model toward producing higher-quality reasoning.

**GRPO** is employed to assess the quality of multimodal CoT reasoning via rule-based reward functions, which are both accurate and robust. For each query, GRPO draws multiple samples and compares the relative quality of the resulting samples, iteratively nudging the model toward higher-quality reasoning segments and thereby improving its capabilities (Guo et al., 2025a; Shao et al., 2024). We follow the standard GRPO framework while incorporating several new practical tricks to enhance training efficiency and stability, as detailed in prior works (Yu et al., 2025). A full description of GRPO is provided in Appendix A.1.

**Rule-Based Reward** is the primary foundation for providing reward signals to the model; its relative magnitude determines the ranking among CoT samples. We employ the classic Format Reward and Accuracy Reward as follows: (1). *Format reward* ensures the correctness of the model's response structure. Specifically, it requires that the reasoning content be delineated with the correct tags, and that the answers provided in `<Recomend Answer>` and `<Answer>` adhere to the specified requirements. (2). *Accuracy reward* evaluates the factual correctness of the model's reasoning. It consists of both per-dimension judgments and an overall preference. An important underlying assumption for GRPO's effectiveness is that if the result satisfies the correctness rules, then the corresponding CoT reasoning sample should reflect a high-quality, accurate reasoning process, thereby truly incentivizing the desired reasoning trajectory.

In conventional RM training, accuracy is assessed only by whether the correct preference is chosen, where the answer space is limited to just three options: `former`, `latter`, and `tie` (Wang et al., 2025b;c). This contradicts our assumption, since many trajectories may have suboptimal multimodal reasoning and insufficient factual grounding yet still produce the correct final answer. Such cases introduce misleading reward signals, reducing efficiency and steering learning in the wrong direction, which harms stability. In contrast, we incorporate both per-dimension judgments and the overall preference. This expands the answer space to $3^{d+1}$, where $d$ is the number of dimensions. For more on sampling efficiency and answer space analysis, please refer to Appendix A.

Formally, the accuracy reward can be written as:

$$r_{\text{acc}} = \alpha \cdot r_{\text{acc\_all}} + \bar{\alpha} \cdot r_{\text{acc\_dim}}, \text{ where } \alpha + \bar{\alpha} = 1,$$

$$r_{\text{acc\_all}} = \mathbb{1}(J_{\text{all}} = \hat{J}_{\text{all}}), r_{\text{acc\_dim}} = \frac{1}{d} \sum_{i=1}^{d} \mathbb{1}(J_{\text{dim\_i}} = \hat{J}_{\text{dim\_i}}).$$

where $J_{\text{all}}$ is the overall judgment, $J_{\text{dim\_i}}$ is the judgment for the $i$-th dimension, and $\hat{J}_{\text{all}}, \hat{J}_{\text{dim\_i}}$ denote the respective ground truths. The function $\mathbb{1}(\cdot)$ is an indicator function that returns 1 if the condition is true and 0 otherwise. $\alpha$ is a tunable hyperparameter that controls the relative importance of the overall preference and the per-dimension judgment.

**CoT Gain Reward** is designed to reward the improvement in accuracy brought by the updated answers in each reasoning segment. This reward is intended to encourage the model to obtain more visual evidence through visual reasoning, update its conclusions with greater accuracy and factual alignment, and thereby strengthen its visual reasoning abilities:

$$r_{\text{cot}} = k \cdot \left( \sum_{i=1}^{t-1} \Delta r_i \right),$$

where $\Delta r_i = r_{acc}^{i+1} - r_{acc}^{i}$ represents the improvement in the accuracy reward between successive updates in the reasoning chain. Here, $i$ denotes the $i$-th reasoning step, $t$ is the total number of reasoning steps, and $k$ is a hyperparameter used to control the degree of the reward.

**Exploratory Incentive** is designed to prevent the model from defaulting to textual reasoning, which can reduce or even degrade its visual reasoning capabilities (Su et al., 2025a). As stated earlier, VLMs inherently possess stronger textual reasoning abilities compared to visual reasoning. During the GRPO process, two factors exacerbate this issue: first, errors in visual tool invocation can lead to negative rewards; second, a certain proportion of queries can achieve decent results through purely textual reasoning, making it difficult for the model to overcome a local optimum .

To encourage exploration, we enforce a lower bound on the proportion of multimodal reasoning produced by the model. This turns the RL objective into a constrained optimization problem, which can be converted into an unconstrained one via Lagrangian Relaxation, as detailed in Appendix A. Formally, the transformed objective can be viewed as adding an auxiliary exploratory reward $r_{\text{explo}}$:

$$r_{\text{explo}} = \max(\omega - \mathcal{R}(\mathbf{X}), 0) \cdot \mathbf{1}_{\text{mul}}(\mathbf{R}),$$

where $\omega$ represents the lower bound on the proportion, $\mathcal{R}(\mathbf{X})$ denotes the proportion of multimodal reasoning in the samples for the query $\mathbf{X}$, and $\mathbf{1}_{\text{mul}}(\cdot)$ is an indicator function that determines whether the sample $\mathbf{R}$ corresponds to multimodal reasoning.

## 4 EXPERIMENTS

### 4.1 EXPERIMENTAL SETUP

**Datasets.** For *training*, we use three datasets: VideoGen-Reward (182k) (Liu et al., 2025b), MJ-Bench-Video (train) (8.7k) (Tong et al., 2025), and Text2Video-Human Preferences (2.6k) by Rapidata[1]. In addition, we distill 1.2k high-quality Multimodal CoT Cold Start samples from GPT-4o (Hurst et al., 2024); these are randomly drawn in proportion from a blend of the three training datasets, and the corresponding original samples are excluded from subsequent training stages. For *benchmarking*, we evaluate on the video part of GenAI-Bench (Jiang et al., 2024), VideoGen-RewardBench (Liu et al., 2025b), and MJ-Bench-Video (test) (Tong et al., 2025). More details on dataset processing and settings are provided in Appendix B. **Base Model.** As a VLM-based reward model, VideoSearch Reasoner is initialized from Qwen2.5-VL-7B (Bai et al., 2025), which has strong visual understanding and video temporal perception capabilities. This provides a solid foundation for unlocking the model's multimodal reasoning potential in long-video scenarios. **Benchmarking.** We compare multiple baseline reward models and VIDEOSEARCH REASONER using greedy decoding across the aforementioned video preference benchmarks. These benchmarks span a wide range of topics and originate from various video generation models (Liu et al., 2025b; Tong et al., 2025; Jiang et al., 2024), ensuring generality of evaluation. We provide detailed descriptions of the baseline models and benchmark datasets in Appendix B. For more detail, please refer to our code at https://anonymous.4open.science/r/videosearchreasoner/.

### 4.2 MAIN RESULTS

Table 1 compares VIDEOSEARCH REASONER against a range of high-performing reward models. Across both evaluation protocols, tau (which accounts for ties) and diff (which excludes ties), our model achieves state-of-the-art performance, significantly surpassing both classic classifier-based and generative-based models, with an average improvement of up to 11.4%. It also outperforms emerging reasoning-style models, owing to our model cultivating not only textual reasoning but also visual reasoning capabilities; when datasets contain more frames than the preset input limit, typical RMs that rely on downsampling inevitably miss key information, whereas our model achieves higher accuracy by processing frames without predetermined limits. Moreover, compared with UNIFIEDREWARD and UNIFIEDREWARD-THINK (Wang et al., 2025b;c), which are both trained on multiple tasks spanning image and video datasets to obtain substantial mutual benefits, our model is trained purely on video preference datasets, yet still surpasses these mutual benefits. These results provide strong evidence for the effectiveness and superiority of our Thinking-with-Image framework, which shows the positive impact of multimodal reasoning for reward models. For further experiments, please refer to the additional experiments section in Appendix C.

---

[1]https://huggingface.co/datasets/Rapidata

**Table 1:** Preference accuracy on evaluation dataset. `tau`: accuracy is calculated with ties included; `diff` excludes tied pairs when calculating accuracy. Best performance in **Bold**.

| Model | Size | GenAI-Bench | | VideoGen-Reward | | MJBench-Video | |
|---|---|---|---|---|---|---|---|
| Protocol | | tau ↑ (%) | diff ↑ (%) | tau ↑ (%) | diff ↑ (%) | tau ↑ (%) | diff ↑ (%) |
| `Classifier-based Reward Models` | | | | | | | |
| VideoScore | 7B | 47.5 | 70.9 | 41.9 | 50.2 | 57.9 | 63.5 |
| VideoReward | 2B | 49.9 | 73.1 | 60.8 | 73.8 | 56.8 | 62.6 |
| VisionReward | 13B | 52.6 | 72.7 | 57.9 | 68.4 | 54.1 | 65.2 |
| `Generative-based Reward Models` | | | | | | | |
| LiFT | 13B | 38.1 | 59.4 | 40.1 | 57.9 | 42.5 | 51.4 |
| UnifiedReward | 7B | 61.2 | 76.8 | 67.1 | 78.6 | 63.3 | 69.5 |
| `Reasoning-based Reward Models` | | | | | | | |
| UnifiedReward-Think | 7B | 64.7 | 80.4 | 69.7 | 79.1 | 62.8 | 71.9 |
| VIDEOSEARCH REASONER | 7B | **68.7** | **82.3** | **71.8** | **80.5** | **67.3** | **75.6** |

## 4.3 ABLATION STUDIES

**Ablation of Visual Reasoning** In our VIDEOSEARCH REASONER framework, we perform tool invocation via Thinking with Image to retrieve visual information and enable multimodal reasoning. To assess the effectiveness of visual reasoning within each reasoning segment, we conduct an ablation on the usefulness of retrieved visual information during tool invocation. Specifically, we compare retrieval guided by the model's visual reasoning–driven tool invocations against randomly retrieving information from the same video data regardless of the tool invocation. As shown in Figure 3, the random strategy yields a clear performance drop, demonstrating that visual reasoning is indispensable for discovering the additional visual evidence needed for reliable judgments.

**Ablation of Training pipeline** We adopt a multi-stage training pipeline and hence conduct ablations on each stage. Following prior work on reasoning-based general models and reward models(Wang et al., 2025b; Guo et al., 2025a), our ablations center on GRPO-based reinforcement fine-tuning, comparing the gains from the cold-start and Rejection sampling Fine-Tuning stages on the final GRPO-trained model. As shown in Figure 3, GRPO contributes the most substantial performance improvement, while both cold start and Rejection sampling Fine-Tuning provide crucial reasoning foundations that further boost post-GRPO performance. Notably, the gains from Rejection sampling Fine-Tuning are especially pronounced, likely because it increases the likelihood of high-quality reasoning segments, thereby improving the efficiency of GRPO-driven improvements.

**Ablation of Auxiliary Reward Setting** In the GRPO stage, we augment the standard format and rule-based accuracy rewards (Shao et al., 2024) with several auxiliary rewards. We conduct ablation studies to quantify the impact of these auxiliary rewards, with results shown in Figure 3. We observe clear performance drops when removing the CoT gain reward and the exploratory incentive. Notably, removing the CoT gain reward has a more pronounced negative effect, highlighting its importance in encouraging the reward model to attempt multimodal reasoning.

**Ablation of Different Accuracy Reward Signals.** In the GRPO stage, beyond the auxiliary rewards described above, we specially design the accuracy reward as a linear combination of the overall reward and per-dimension reward to enlarge the answer space. We conduct ablations to assess their effects, comparing three settings: using only the overall reward, using only the per-dimension reward, and using a 50/50 mix of overall and per-dimension rewards (the setting we adopt). The results, shown in Figure 3, validate the benefits of the mixed scheme.

## 4.4 FURTHER ANALYSIS

**Visualization on GRPO Training** For a deeper analysis of the GRPO stage and the differences in training under various baselines, we provide a visualization of GRPO training in Figure 4. It highlights the model's changes in evaluation accuracy, average number of tool invocations per sample, and average length per reasoning segment in different experimental settings, including: setting of VIDEOSEARCH REASONER, without exploratory reward, without per-dimension accuracy reward ($\alpha = 1$), and without overall accuracy reward ($\alpha = 0$).

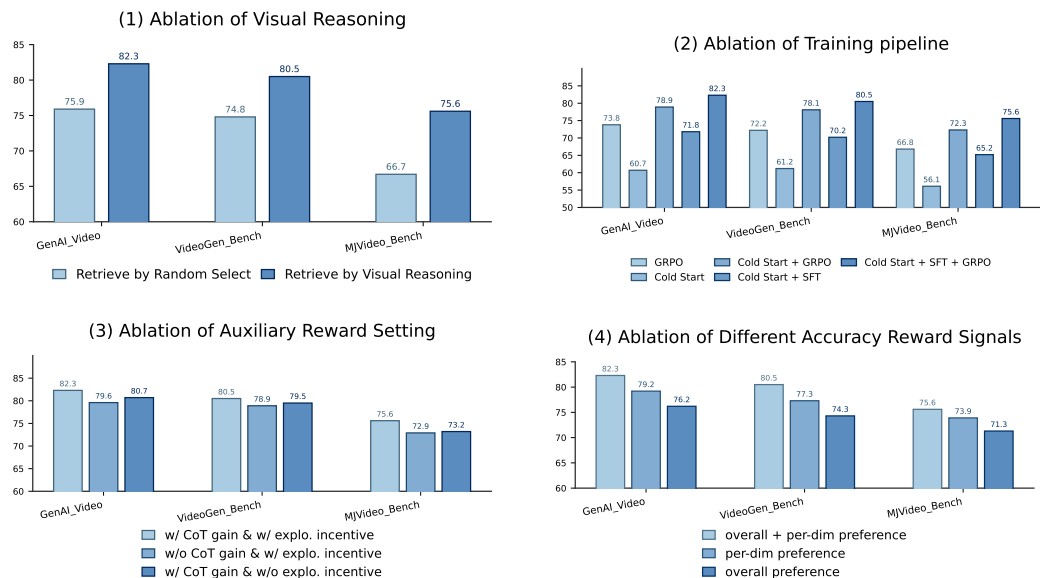

**Figure 3:** The results of ablation studies are summarized in this figure: **(1)** investigates the ablation of **visual reasoning**; **(2)** examines the impact of different **training stages** on the final model performance; **(3)** explores ablations of different **auxiliary reward** settings; and **(4)** studies the ablation of different **accuracy reward** signals by our modification of the accuracy reward.

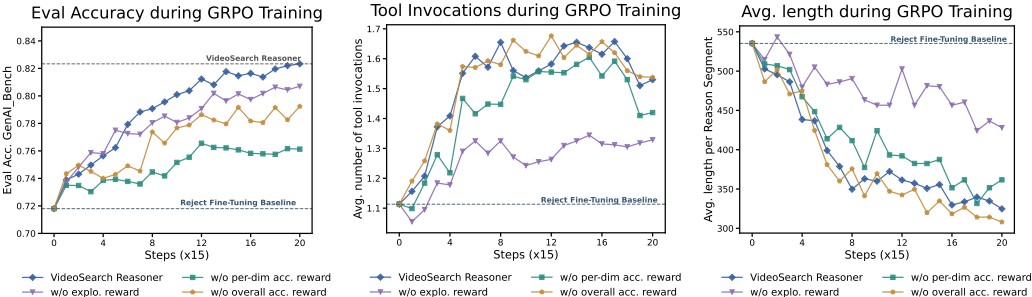

**Figure 4:** The training dynamics of the GRPO stage: **(1)** accuracy on GenAI-Bench throughout training; **(2)** average tool invocations per sample; **(3)** average reasoning segment length.

**Error Analysis**   To more rigorously validate that our RM on long videos and complex reasoning scenarios, we conduct an error analysis. Standard video preference datasets comprise videos of varying lengths produced by multiple generators and prompted at different complexity levels. For instance, in VideoGen-RewardBench, 16.4% of videos contain roughly 49 frames, whereas 15.7% contain approximately 173 frames, resulting in a 3.5× disparity. Shorter videos are typically easier for baseline models, obscuring our advantage in visual reasoning, while higher prompt complexity further increases content richness and alignment demands, thereby making RM evaluation more challenging. To better assess our model under these difficult scenarios, especially in comparison to native generative outputs and text-only reasoning paradigms (namely, LIFT, UNIFIEDREWARD, and UNIFIEDREWARD-THINK), we perform a secondary filtering of each dataset to construct two "hard" subsets by selecting the top 10% by video length and the top 10% by prompt length. Results are reported in Table 2. It can be seen that, compared with baseline models, VIDEOSEARCH REASONER shows a smaller drop in accuracy on all of the hard subsets.

## 5 CONCLUSION

In this work, we introduce VideoSearch Reasoner, the first multimodal RM capable of visual reasoning. VideoSearch Reasoner leverages the Thinking-with-Image framework to alleviate context-length constraints and mitigate forgetting of visual information. We adopt a three-stage training pipeline to progressively enhance both textual and visual reasoning abilities. Extensive experiments

**Table 2:** Preference accuracy on Long Video and Complex Prompt subset. **tau**: accuracy is calculated with ties included; **diff** excludes tied pairs when calculating accuracy. Best performance in **Bold**.

| | | Long Video | | | | | |
|---|---|---|---|---|---|---|---|
| Model | Size | GenAI-Bench (long) | | VideoGen-Reward (long) | | MJBench-Video (long) | |
| Protocol | | tau ↑ (%) | diff ↑ (%) | tau ↑ (%) | diff ↑ (%) | tau ↑ (%) | diff ↑ (%) |
| LiFT | 13B | 36.0 | 56.5 | 35.8 | 53.6 | 39.5 | 50.4 |
| UnifiedReward | 7B | 56.8 | 71.6 | 63.5 | 72.2 | 59.6 | 67.3 |
| UnifiedReward-Think | 7B | 61.7 | 76.4 | 65.8 | 76.7 | 60.1 | 69.6 |
| **VIDEOSEARCH REASONER** | 7B | **66.2** | **81.4** | **70.9** | **79.6** | **66.1** | **74.8** |
| | | Complex Prompt | | | | | |
| Model | Size | GenAI-Bench (complex) | | VideoGen-Reward (complex) | | MJBench-Video (complex) | |
| Protocol | | tau ↑ (%) | diff ↑ (%) | tau ↑ (%) | diff ↑ (%) | tau ↑ (%) | diff ↑ (%) |
| LiFT | 13B | 37.6 | 58.7 | 40.5 | 57.6 | 39.8 | 50.8 |
| UnifiedReward | 7B | 58.8 | 74.9 | 65.2 | 76.6 | 62.4 | 69.1 |
| UnifiedReward-Think | 7B | 63.9 | 79.8 | 68.2 | 78.2 | 60.5 | 70.1 |
| **VIDEOSEARCH REASONER** | 7B | **68.4** | **81.9** | **70.6** | **80.7** | **66.3** | **74.3** |

shows the effect of our framework, which improves the accuracy of preference judgments and the interpretability of reward signals.

## ETHICAL STATEMENTS

This work fully adheres to the ICLR Code of Ethics in all aspects of research conduct. The data collection, usage, and model development processes comply with ethical standards regarding privacy, consent, and responsible AI practices. To the best of our knowledge, this study does not involve any data, methodologies, or applications that raise ethical concerns. The authors confirm that they have reviewed and followed the ICLR Code of Ethics throughout this research.

## REPRODUCIBILITY STATEMENT

To ensure the reproducibility of our work, we have made the following comprehensive efforts. First, we provide detailed descriptions of the data construction pipeline and input prompt templates in Section 3.1 and Appendix D. Second, we present thorough documentation of the processing templates and workflows for handling video-query pairs from datasets as inputs in Appendix D. Third, as described in Section 4.1, we have open-sourced our training code for each stage of the training process, which is available at https://anonymous.4open.science/r/videosearchreasoner/. Fourth, for all theoretical results and corresponding insights claimed in this paper, we provide complete proofs and explanations in Appendix A. These efforts, combined with detailed descriptions throughout the paper, fully guarantee the reproducibility of our research findings and enable other researchers to validate and build upon our work.

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

# APPENDIX

The appendix of this paper is organized as follows: Appendix A provides mathematical details and derivations omitted from the main text; Appendix B supplements additional experimental details; Appendix C presents more extensive experimental results; Appendix D includes prompt templates. Appendix E provides further discussion on failure modes and terminology. Appendix F and G will describe the limitations and the LLM usage, respectively.

## A MATHEMATICAL ANALYSIS

### A.1 MATHEMATICAL DETAILS OF THE TRAINING PIPELINE

**Supervised fine-tuning (SFT) loss.** As mentioned in Section 3.2.1, our training comprises two major stages: Cold Start and Supervised Fine-Tuning. For high-quality CoT data constructed via the specific pipeline, we use the standard supervised fine-tuning loss while masking tokens associated with tool-execution outcomes from the loss computation. Formally, in the multi–reasoning-segment setting, the SFT loss is:

$$\mathcal{L}_{\text{sft}}(\theta) = -\sum_{i=1}^{t} \sum_{j=1}^{N_i} \log p\left(r_{i,j} \mid \mathcal{X}, (r_1, o_1), \dots, (r_{i-1}, o_{i-1}), r_{i,<j}; \theta\right), \tag{1}$$

where $\theta$ denotes the parameters of the reward model (RM), $\mathcal{X} = [\mathcal{V}, T]$ represents the pair of the initial visual input $\mathcal{V}$ and the query template $T$, $r_i$ is the $i$-th reasoning segment, $r_{i,j}$ is the $j$-th token of the $i$-th reasoning segment, $o_i$ is the $i$-th tool-execution outcome, $N_i$ is the total number of tokens in the $i$-th reasoning segment, and $t$ is the total number of CoT steps.

**GRPO Algorithm.** As mentioned in Section 3.2.2, GRPO-based reinforcement fine-tuning is employed because the rule-based reward function provides a robust reward signal to nudge the model toward generating higher-quality reasoning segments. The specific algorithm is similar to the one described in Shao et al. (2024), with some novel practical tricks introduced in Yu et al. (2025).

For each input $\mathcal{X} = [\mathcal{V}, T]$ (the pair of the initial visual input $\mathcal{V}$ and the query template $T$), a set of CoT samples is randomly drawn from the same model $\pi_\theta(\cdot)$, denoted as $G = \{\tilde{R}_{1,t_1}, \dots, \tilde{R}_{n,t_n}\}$, where $n$ refers to the number of sampled CoT examples, and $R_{i,t_i}$ represents the $i$-th CoT sample with $t_i$ reasoning segments.

A predefined reward function $f(\cdot) = \sum_i f_i(\cdot)$ is applied to each sample, resulting in $S = \{\sum_i f_i(R_{1,t_1}), \dots, \sum_i f_i(R_{n,t_n})\} = \{s_1, \dots, s_n\}$, where the specific $f(\cdot)$ in our setting is defined as:

$$f(\cdot) = f_{\text{fmt}}(\cdot) + f_{\text{acc}}(\cdot) + f_{\text{cot}}(\cdot) + \eta f_{\text{explo}}(\cdot),$$

where $\beta$ and $\eta$ are adjustable hyperparameters, predefined here for simplicity. This is followed by intra-group normalization to calculate the advantage for each sample: $\mathcal{A}_i = \{s_i - \mu(S)\}/\sigma(S)$, where $\mu(S)$ represents the mean of the scores in the set $S$ and $\sigma(S)$ represents the standard deviation of the scores in the set $S$.

Subsequently, the likelihood ratio of each response is computed to guide the model toward higher-quality reasoning segments:

$$\zeta_{i,t} = \frac{\pi_\theta(r_{i,t} \mid \mathcal{X}, (r_1, o_1), \dots, (r_{i-1}, o_{i-1}), r_{i,<t})}{\pi_{\theta_{\text{old}}}(r_{i,t} \mid \mathcal{X}, (r_1, o_1), \dots, (r_{i-1}, o_{i-1}), r_{i,<t})},$$

where $\pi_\theta$ represents the new policy and $\pi_{\theta_{\text{old}}}$ represents the old policy.

The final optimization objective in GRPO is:

$$\mathcal{J}_{\text{grpo}}(\theta) =$$

$$\mathbb{E}_{[\mathcal{X} \sim \mathcal{D}, \tilde{R}_{i,t_i} \sim \pi_{\theta_{\text{old}}}]} \frac{1}{\mathcal{T}(\tilde{R}_{i,t_i})} \sum_{t=1}^{\mathcal{T}(\tilde{R}_{i,t_i})} \left\{ \left[ \min\left(\zeta_{i,t}, \texttt{clip}(\zeta_{i,t}, , 1-\varepsilon, 1+\varepsilon)\right) \mathcal{A}_i \right] - \beta \mathbb{D}_{\text{KL}}[\pi_\theta \parallel \pi_{\text{ref}}] \right\}$$

where $\mathcal{D}$ represents the dataset, $\mathcal{T}(\tilde{R}_{i,t_i})$ denotes the total number of tokens in the multimodal CoT sample, clipping within $1 - \varepsilon$ ensures training stability, and $\mathbb{D}_{\mathrm{KL}}$ is the KL divergence penalty to constrain the model update range.

As previously studied in Yu et al. (2025), we incorporate a **Dynamic Sampling** improvement into our GRPO training algorithm. Specifically, when drawing a batch of samples, if the accuracy is 1 or 0, the entire batch's advantage becomes zero, yielding zero gradients for that batch. This effectively reduces the gradient-accumulation batch size, increases noise sensitivity, and lowers sample efficiency. The issue worsens as training progresses and accuracy rises, since fully correct cases become more frequent, leading to more zero-gradient batches. Dynamic Sampling mitigates this by filtering out batches whose accuracy is 1 or 0 and resampling until all batches yield nonzero gradients, thereby improving training efficiency.

**Sampling efficiency and answer-space in GRPO.** We first analyze, as in Section 3.2.2, how the size of the answer space affects GRPO sampling and learning efficiency. Let the answer space size be $N$, the observed model accuracy be $p$, the model's intrinsic accuracy be $q$ (interpreted as "finding the key information correctly and thus making the correct judgment"), and the proportion of invalid samples be $r$ (failing to find the key information, yet coincidentally producing the correct judgment). We have:

$$p = q + (1 - q)/N, \tag{1}$$
$$r = (1 - q)/(N) = (1 - p)/(N - 1). \tag{2}$$

For the $(1 - q)$ fraction of samples where key information is not found, the model's judgment can be viewed as randomly selecting an answer from an answer space of size $N$, which yields an additional accuracy of $(1 - q)/N$, giving Equation (1). For Equation (2), although these $(1 - q)/N$ samples happen to produce correct judgments, their reasoning lacks the key information and is thus off-point; we term them invalid samples. In reinforcement learning (RL), assigning these samples high advantage and increasing their likelihood is not only unhelpful for improving the model, but can be harmful. The expression $(1 - p)/(N - 1)$ thus provides an estimate of the proportion of such invalid samples.

Take the observed accuracy $p$ as an intermediate value during training, say $0.7$. Then: For $N = 3$ (setting in classic RM training), the estimated invalid data proportion is $r = (1 - 0.7)/2 = \mathbf{15\%}$. For $N = 3^{d+1} = 81$ (our setting with $d = 3$), the estimated invalid data proportion is $r = (1 - 0.7)/80 = \mathbf{0.375\%}$, which greatly reduces the fraction of invalid data and improves sampling effectiveness.

Next, we analyze the impact of accuracy $p$ in Dynamic Sampling, as stated in A.1. Denote the batch sample size by $n$. The probability that a batch is entirely correct or entirely wrong is:

$$r' = p^n + (1 - p)^n.$$

Taking $p = 0.7$ and $n = 8$, we get:

$$r' = 0.7^8 + (1 - 0.7)^8 = \mathbf{16.7\%}.$$

Without a Dynamic Sampling mechanism, this nontrivial fraction of ineffective batches would indeed hamper training.

## A.2 DERIVATION OF THE GRPO EXPLORATORY INCENTIVE

Here, we provide a more detailed explanation of the design and derivation of the Exploratory Incentive. The reason the Exploratory Incentive is not directly designed as an auxiliary reward that increases according to the multimodal CoT ratio $\mathcal{R}$, which would be simpler, is because merely adding rewards may lead to reward hacking. In such cases, the model might excessively prioritize generating visual CoTs, resulting in useless reasoning that hinders the development of well-integrated multimodal reasoning capabilities. Inspired by Su et al. (2025a), we transform this problem into a constrained optimization problem. This ensures that the final optimization objective does not explicitly contain the multimodal CoT ratio $\mathcal{R}$, thereby avoiding the issue of reward hacking. Meanwhile, by incorporating the multimodal CoT ratio $\mathcal{R}$ into the constraints, we achieve the goal of preventing degeneration and maintaining the desired behavior.

Formally, the original reinforcement learning problem is an unconstrained optimization problem, written as:

$$\max_\theta \quad \mathbb{E}\left[r(\mathcal{X}, \tilde{R}_t) \mid \mathcal{X} \sim \mathcal{D}, \tilde{R}_t \sim \pi_\theta(\cdot \mid \mathcal{X})\right],$$

where $r(\mathcal{X}, \tilde{R}_t)$ represents the reward, $\mathcal{X}$ is the input sampled from the dataset $\mathcal{D}$, and $\tilde{R}_t$ is the CoT sample with $t$ reasoning steps generated by the policy $\pi_\theta(\cdot \mid \mathcal{X})$.

After adding constraints, the optimization problem becomes a constrained one:

$$\max_\theta \quad \mathbb{E}\left[r(\mathcal{X}, \tilde{R}_t) \mid \mathcal{X} \sim \mathcal{D}, \tilde{R}_t \sim \pi_\theta(\cdot \mid \mathcal{X})\right] \tag{2}$$

$$\text{subject to,} \quad \mathcal{R}(\mathcal{X}) \geq \omega \tag{3}$$

Where $\mathcal{R}(\mathbf{X})$ denotes the proportion of multimodal reasoning in the samples for the query. The constraint can be rewritten as $g(\mathcal{X}, \theta) = \omega - \mathcal{R}(\mathcal{X}) \leq 0$. We apply the Lagrangian Relaxation method (**?**) to incorporate this constraint into the optimization objective. Unlike the standard Lagrangian method, which rewrites the objective as:

$$r_{new}(\mathcal{X}, \tilde{R}_t) = r(\mathcal{X}, \tilde{R}_t) - \lambda \cdot (\omega - \mathcal{R}(\mathcal{X})),$$

where $\lambda \geq 0$ is the Lagrange multiplier, we instead follow the approach described in Su et al. (2025a); Wang et al. (2022), which uses the formulation:

$$r_{new}(\mathcal{X}, \tilde{R}_t) = r(\mathcal{X}, \tilde{R}_t) + \eta \cdot \max(\omega - \mathcal{R}(\mathcal{X}), 0) \cdot \mathbf{1}_{\text{mul}}(\tilde{R}_t),$$

where $\eta \geq 0$ is a fixed hyperparameter.

This formulation preserves equivalence to the original constrained objective while offering significant benefits during GRPO: unlike standard Lagrangian methods, where the multiplier $\lambda$ needs to be dynamically adjusted, as derived in Wang et al. (2022), this structure avoids that requirement. Instead, it allows $\eta$ to be treated as a fixed hyperparameter. By pre-selecting $\eta$, this transformation can then be interpreted during RL training as adding an additional exploratory incentive reward, making the computation highly convenient:

$$r_{\text{expo}} = \max(\omega - \mathcal{R}(\mathcal{X}), 0) \cdot \mathbf{1}_{\text{mul}}(\tilde{R}_t).$$

# B   DETAILED EXPERIMENTAL SETTINGS

## B.1   TRAINING DETAILS.

**Pipeline details.**   For the cold start and Rejection sampling Fine-Tuning data, we referenced and modified the TRL code. For CoT samples, we compute the SFT loss (as stated in A.1) with a batch size of 1 and set gradient accumulation steps to 32. For the GRPO stage, we adopt and adapt the OpenRLHF training code. In each batch, the number of queries is set to 64, and the number of responses per query $N$ is set to 8; accordingly, the samples collected per training batch total 512. We update the behavior policy model with the improved policy model every 4 batches, corresponding to experience from 256 queries. 8 NVIDIA A800 (80GB) GPUs are used for both the cold start and Rejection sampling Fine-Tuning stages, while 32 NVIDIA A800 (80GB) GPUs are used for the GRPO stage.

**Hyperparameters.**   For cold start and Rejection sampling Fine-Tuning, we use a learning rate of $1.5 \times 10^{-6}$ with a warm-up ratio of 0.2. During the GRPO stage, we use a learning rate of $10^{-6}$ with a KL penalty coefficient of $\beta = 0.01$. Additionally, for reward-related hyperparameters: $\alpha$, which controls the balance between per-dimension and overall preference in the accuracy reward, is set to 0.5, selected via parameter search. The parameter $k$, which controls the strength of the CoT gain reward, is set as 0.2 to balance emphasizing visual reasoning and avoiding excessive strength that could cause reward hacking (see Appendix C.6 for detailed analysis). For $\eta$, the hyperparameter governing the exploratory incentive reward as detailed in Appendix A.2, we set it to 0.5; correspondingly, the minimum multimodal reasoning ratio in the constraint, $\omega$, is set to 0.2. For the window width $p$, we default to 1, considering GPU memory limitations and the `<Snapshot>` mechanism's preservation of salient information.

## B.2 COMPARED BASELINES.

We compare our model against a range of leading, high-performing reward models. We categorize the compared models into three major classes: classifier-based reward models, generative-based reward models, and reasoning-based reward models.

**Classifier-based Reward Models**. These methods build on VLMs but replace the final linear layer of the VLM's LLM backbone. Instead of outputting a next-token distribution, they retrain a linear head to directly produce per-dimension or overall scores (or preferences). In this paradigm, the RMs include **VideoScore** (He et al., 2024), **VisionReward** (Xu et al., 2024), and **VideoReward** (Liu et al., 2025b). They leverage VLMs' strong capabilities for understanding and embedding visual information to produce preference judgments in a single classifier step. While the risk of reward hacking has been highlighted when aligning preferences with such models, such RMs that directly judge visual information still provide strong baselines.

**Generative-based Reward Models**. These models leverage the VLM's intrinsic understanding and generating ability without modifying the model; instead, they treat preference decisions as a visual-language task. By using prompt templates, they tap into the VLM's comprehension and generative capabilities to produce responses and preference judgments. Representative RMs in this paradigm include **LiFT-Critic** (Wang et al., 2024) and **UnifiedReward** (Wang et al., 2025c), which, even without eliciting reasoning, fully leverage VLMs' vision–language alignment and serve as strong baselines.

**Reasoning-based Reward Models**. This emerging class recognizes the close relationship between preference judgment and reasoning, and the positive impact of logical reasoning on producing more accurate outcomes. Models in this category include **UnifiedReward-Think** Wang et al. (2025b), which, via RL-centric training pipelines, elicits the model's textual reasoning ability, yielding strong reasoning-driven baselines that exploit VLMs. Our newly proposed **VIDEOSEARCH REASONER** also belongs to this category but further introduces multimodal reasoning, breaking the VLM's inherent processed-frame limitation and reducing risks of forgetting induced by purely textual reasoning.

## B.3 DATASETS AND USAGE SETTINGS

**Training data setup.** As noted in Section 3.2, we compute the accuracy reward using both per-dimension and overall preferences, which our ablation shows to be crucial. This requires datasets annotated with per-dimension preferences-something that is non-trivial. Many preference datasets used for training, such as **VideoDPO** (Liu et al., 2024) and **LiFT-HRA** (Wang et al., 2024), provide only an overall preference and thus are not usable for our reward design. We therefore select fine-grained datasets with per-dimension labels: **VideoGen-Reward** (182k) (Liu et al., 2025b), **MJ-Bench-Video (train)** (8.7k) (Tong et al., 2025), and **Text2Video-Human Preferences** (2.6k) by Rapidata [2].

Due to differing annotation schemes and label contents, we still need to harmonize fine-grained annotations across datasets: **Dimension selection.** MJ-Bench-Video (train) includes 5 high-level preferences and up to 28 fine-grained preferences. We align its dimensionality with VideoGen-Reward and Text2Video-Human Preferences by selecting three core dimensions: Alignment, Fineness, and Coherence & Consistency. **Dimension semantics.** Since dimension titles differ across datasets, we take two steps: **(i)** For each dataset, we include a dataset-specific explanation in the prompt that clarifies the meaning of each dimension as detailed in Appendix D. **(ii)** We map dimensions with different names but similar semantics to a common triad: VideoGen-Reward's Text Alignment, Visual Quality, and Motion Quality; MJ-Bench-Video's Alignment, Fineness, and Coherence & Consistency; and Rapidata's Text2Video-Human Preferences' Alignment, Preference [3], and Coherence. Although the labels differ in name, they consistently target: (1) alignment to the prompt, (2) intrinsic visual quality, and (3) temporal coherence/motion. This allows the model to learn the underlying correspondences without being misled by naming differences, projecting knowledge onto these three core dimensions.

---

[2]https://huggingface.co/datasets/Rapidata

[3]as per Rapidata, this reflects visual appeal rather than overall preference

**Benchmarking data setup.** As noted above, we evaluate on three high-quality video preference datasets, **GenAI-Bench** (Jiang et al., 2024), **VideoGen-RewardBench** (Liu et al., 2025b), and **MJ-Bench-Video** (Tong et al., 2025), which also serve as mainstream leaderboards for video preference (Wang et al., 2025b). Each dataset contains entries which consist of a prompt, a pair of videos generated from the same prompt (by different models or by different seeds of the same model), and human expert annotations of preference, including an overall preference and, in some cases, per-dimension preferences. For example, VideoGen-RewardBench includes three additional per-dimension metrics: Text Alignment, Video Quality, and Movement Quality; MJ-Bench-Video includes five high-level categories and up to 28 fine-grained preferences; GenAI-Bench provides only an overall preference. To align evaluation with both the leaderboards and our training setup, we keep the same prompt template and required response format as in training, but when computing evaluation accuracy, we use only the model's predicted overall preference. For more detail, please refer to our code at `https://anonymous.4open.science/r/videosearchreasoner/`.

## C  FURTHER EXPERIMENTAL RESULTS

In this section, we present more detailed experimental results, including: **(i)** comparisons of different hyperparameter settings; **(ii)** the effect of varying the amount of rejected fine-tuning data on the GRPO stage; **(iii)** benchmarking performance after excluding hard subsets from the evaluation set and increasing numbers of frames per video; **(iv)** inference latency and throughput benchmarks compared with baselines; **(v)** out-of-distribution (OOD) testing of the model; and **(vi)** evaluation of the model's performance with different cold-start data source.

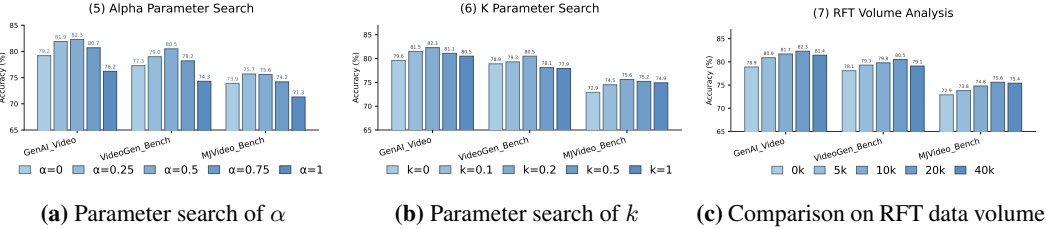

(a) Parameter search of $\alpha$     (b) Parameter search of $k$     (c) Comparison on RFT data volume

**Figure 5:** The results of the hyperparameter search and the reject fine-tuning data volume comparison are summarized in this figure: **(a)** shows parameter search for $\alpha$; **(b)** shows parameter search for $k$; **(c)** shows comparison across rejection sampling fine-tuning data volumes.

### C.1  COMPARISON OF DIFFERENT HYPERPARAMETER CHOICES

To identify the optimal hyperparameters in Appendix B.1, we conducted a parameter search. Specifically, we tuned $\alpha$, which balances the weights of overall accuracy versus per-dimension accuracy, and $k$, which controls the strength of the Chain-of-Thought (CoT) gain reward. The final evaluations are reported in Figure 5a and 5b. We observe that $\alpha$ has a pronounced effect on performance: $\alpha = 1$ reduces to training without the per-dimension accuracy reward, whereas $\alpha = 0$ removes the overall accuracy reward. Our chosen setting, $\alpha = 0.5$, yields the best results. The choice of $k$ also matters, with $k = 0.2$ performing best, indicating that a sufficiently strong CoT gain reward is important. However, larger $k$ values do not further improve performance, likely because the model can game the signal by remaining deliberately neutral in early reasoning steps to secure larger subsequent gains (i.e., reward hacking).

### C.2  COMPARISON OF REJECT FINE-TUNING DATA VOLUME

As shown in Section 3.2, the rejection sampling fine-tuning stage is crucial for consolidating the model's reasoning ability, thereby paving the way for improved GRPO. We further investigate the effect of data volume during the rejection sampling fine-tuning stage for post-GRPO performance; results are presented in Figure 5c. We observe a clear positive correlation of post-GRPO performance and rejection sampling fine-tuning data volume at smaller scales, which is expected: more sampled reasoning patterns that are filtered for quality and correctness lead to better capabilities.

**Table 3:** Preference accuracy on Residual subset and Redundant version dataset. `tau`: accuracy is calculated with ties included; `diff` excludes tied pairs when calculating accuracy. Best performance in **Bold**

| | | Residual Dataset | | | | | |
|---|---|---|---|---|---|---|---|
| Model | Size | GenAI-Bench (residual) | | VideoGen-Reward (residual) | | MJBench-Video (residual) | |
| Protocol | | tau ↑ (%) | diff ↑ (%) | tau ↑ (%) | diff ↑ (%) | tau ↑ (%) | diff ↑ (%) |
| LiFT | 13B | 38.3 | 59.6 | 40.4 | 58.2 | 42.8 | 51.5 |
| UnifiedReward | 7B | 61.5 | 77.2 | 67.5 | 79.0 | 63.6 | 69.7 |
| UnifiedReward-Think | 7B | 65.0 | 80.7 | 70.0 | 79.3 | 63.1 | 72.1 |
| **VIDEOSEARCH REASONER** | 7B | **68.9** | **82.4** | **71.9** | **80.6** | **67.4** | **75.7** |
| | | Redundant Dataset | | | | | |
| Model | Size | GenAI-Bench (redundant) | | VideoGen-Reward (redundant) | | MJBench-Video (redundant) | |
| Protocol | | tau ↑ (%) | diff ↑ (%) | tau ↑ (%) | diff ↑ (%) | tau ↑ (%) | diff ↑ (%) |
| LiFT | 13B | 36.9 | 57.9 | 38.2 | 55.8 | 40.1 | 50.8 |
| UnifiedReward | 7B | 58.9 | 74.7 | 65.2 | 74.2 | 62.1 | 68.7 |
| UnifiedReward-Think | 7B | 63.4 | 77.9 | 66.8 | 77.3 | 61.8 | 70.8 |
| **VIDEOSEARCH REASONER** | 7B | **67.2** | **81.9** | **71.5** | **79.8** | **66.3** | **75.2** |

However, using even more data (40k in our setting) degrades performance, potentially because extensive supervised fine-tuning reduces output entropy, making subsequent GRPO optimization more difficult.

## C.3    EVALUATION ON REMAINED AND DUPLICATED SET

To better compare improvements across different components of the evaluation set (grouped by prompt complexity and frame count) and assess whether gains are larger on complex scenarios and longer videos, in addition to the results on the Longer video and Complex prompt subsets reported in Table 2, we also report results on the rest of the dataset for comparison. As shown in Table 3, relative to Table 2, the improvements on the Residual subset are less pronounced than on the Longer video and Complex prompt subsets, which validates our analysis.

Beyond direct evaluation on our Video Preference Dataset, we further probe the model's ability to mine and analyze information from long videos by artificially increasing data size. Concretely, we inject redundant visual information by duplicating frames: frames at random positions are duplicated a number of times equal to the original video length, doubling the total frame count. On this redundancy-augmented dataset, results in Table 3 show that our model experiences a smaller performance drop compared with other models.

## C.4    INFERENCE METRICS COMPARISON WITH BASELINE MODELS

Our approach enhances the reward model through multimodal reasoning. Compared with **aggressively downsampled** naive reward models that process only a few frames (e.g. 8 frames inputs), our model involves longer inference chains, indeed resulting in higher latency compared. However, as the demand for more accurate reward signals increases and the generated videos become longer, these naive models must expand their input length to maintain fidelity, driving their computational cost up **significantly**. In contrast, one of the key motivations of our design is to **mitigate this prohibitive cost growth** when scaling to high-quality, long videos. By leveraging a **window memory mechanism** rather than feeding all frames into the context window at once, the GPU HBM footprint remains stable and manageable. This design allows our model to achieve SOTA accuracy at a fraction of the cost required by naive long-context models.

We benchmarked VIDEOSEARCH REASONER against baseline models using varying numbers of input frames (8, 32, and 256) to quantify both accuracy and system efficiency. The results are summarized in Table 4 and  5.

The results demonstrate that our model pushes the **accuracy–efficiency Pareto frontier**. Naively increasing the number of input frames (e.g. Unified-Reward(256)) leads to out-of-memory (OOM) on a single GPU and a severe drop in throughput, with diminishing or even negative returns in accuracy. This is likely due to the model struggling to handle an excessive number of visual tokens and less important information, a known issue in long-context modeling (Wu et al., 2025b).

**Table 4:** Preference accuracy on evaluation dataset. Best performance in **Bold**

| Model | GenAI-Bench diff ↑ (%) | VideoGen-Reward diff ↑ (%) | MJBench-Video diff ↑ (%) |
|---|---|---|---|
| UnifiedReward(8) | 76.8 | 78.6 | 69.5 |
| UnifiedReward(32) | 77.2 | 78.9 | 69.9 |
| UnifiedReward(256) | 76.3 | 78.7 | 70.3 |
| UnifiedReward-think(8) | 80.4 | 79.1 | 71.9 |
| UnifiedReward-think(32) | 81.0 | 79.8 | 72.5 |
| UnifiedReward-think(256) | 79.8 | 78.7 | 73.1 |
| VIDEOSEARCH REASONER | **82.3** | **80.5** | **75.6** |

**Table 5:** Comparison of system performance on A800 GPUs

| Model | KV Cache (GB) | Latency (s) | Speed (tok/s) | Max Concurrency | Throughput (tok/s) |
|---|---|---|---|---|---|
| | 1×A800 (Single GPU) | | | | |
| UnifiedReward(8) | 3.96 | 0.93 | 74 | 16 | 274 |
| UnifiedReward(32) | 15.81 | 2.85 | 44 | 4 | 68 |
| UnifiedReward(256) | OOM | – | – | – | – |
| UnifiedReward-think(8) | 4.09 | 5.40 | 73 | 16 | 266 |
| UnifiedReward-think(32) | 16.11 | 10.31 | 44 | 4 | 67 |
| UnifiedReward-think(256) | OOM | – | – | – | – |
| **VideoSearch Reasoner** | **7.12** | **7.55** | **63** | **9** | **162** |
| | 8×A800 (Multi-GPU) | | | | |
| UnifiedReward(8) | 3.96 | 0.12 | 589 | 158 | 2612 |
| UnifiedReward(32) | 15.81 | 0.36 | 355 | 39 | 654 |
| UnifiedReward(256) | 126.43 | 2.60 | 75 | 4 | 81 |
| UnifiedReward-think(8) | 4.09 | 0.67 | 584 | 152 | 2326 |
| UnifiedReward-think(32) | 16.11 | 1.29 | 351 | 38 | 642 |
| UnifiedReward-think(256) | 128.31 | 7.02 | 74 | 4 | 80 |
| **VideoSearch Reasoner** | **7.12** | **0.92** | **501** | **87** | **1553** |

Moreover, the reduced memory footprint results in clear **system-level benefits**. On a single A800 GPU, our model supports $2.25\times$ higher concurrency (9 vs. 4) and achieves $2.4\times$ greater throughput (162 vs. 67 tok/s) compared to Unified-Reward-think(32), while achieving superior performance. Furthermore, its throughput are even comparable to 8-frame baselines, showcasing an excellent tradeoff between accuracy and efficiency. Therefore, VIDEOSEARCH REASONER delivers strong system-level performance and is well-suited for downstream deployment.

## C.5 OUT-OF-DISTRIBUTION EVALUATION

To further assess the generalization capability of our reward model, we conduct a set of out-of-distribution evaluations across multiple benchmarks. We first clarify the in-distribution (ID) versus out-of-distribution (OOD) settings of our datasets. Although the names of our primary training dataset, **VideoGen-Reward**, and one of our evaluation benchmarks, **VideoGen-RewardBench**, appear similar, they are in fact constructed **independently**, ensuring that the reported performance reflects genuine generalization rather than in-distribution bias. As described in (Liu et al., 2025b) the **VideoGen-Reward** training set (182k pairs) was built from videos generated using 16k prompts and a fixed collection of 12 text-to-video (T2V) models, followed by human annotation. In contrast, the **VideoGen-RewardBench** evaluation dataset was derived from a separate third-party public dataset (*VideoGen-Eval* (Zeng et al., 2024)) and includes videos produced by different T2V models with distinct sampling parameters. Evaluation on VideoGen-RewardBench therefore constitutes a clear **OOD setting** relative to the training distribution.

Similarly, the **GenAI-Bench** benchmark was used solely for evaluation, without including its corresponding training split in our data pipeline, which further eliminates potential in-distribution (ID) contamination. A partial distributional overlap may exist for the **MJ-Bench-Video** dataset, as its training and test subsets were randomly split from the same large dataset (Tong et al., 2025). However, MJ-Bench-Video accounts for only a small fraction (approximately 4%) of our total training data. To quantify the potential influence of this overlap, we conducted an ablation study in which all MJ-Bench-Video data were completely excluded from training. The results are summarized in Table 6.

**Table 6:** Preference accuracy on evaluation datasets under out-of-distribution conditions. `tau`: accuracy is calculated with ties included; `diff` excludes tied pairs when calculating accuracy.

| Model Configuration | GenAI-Bench | | VideoGen-RewardBench | | MJ-Bench-Video | |
|---|---|---|---|---|---|---|
| | tau ↑ (%) | diff ↑ (%) | tau ↑ (%) | diff ↑ (%) | tau ↑ (%) | diff ↑ (%) |
| VIDEOSEARCH REASONER (Full Data) | 71.8 | 80.5 | 68.7 | 82.3 | 67.3 | 75.6 |
| VIDEOSEARCH REASONER (No MJ-Bench) | 71.9 | 80.4 | 68.7 | 82.4 | 66.8 | 75.1 |

The results show that excluding MJ-Bench-Video from the training data has only a negligible influence on the other benchmarks (GenAI-Bench and VideoGen-RewardBench). The slight decrease on the MJ-Bench-Video test set demonstrates that the model has learned generalizable principles of video quality assessment for reward modeling.

### C.6 EFFECT OF COLD-START DATA SOURCE

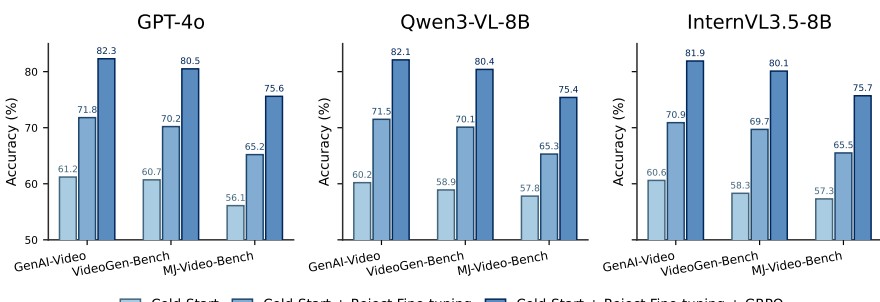

**Figure 6:** Performance comparison across different cold-start data sources.

Our training pipeline is initially guided by "high-quality CoT trajectories" generated by GPT-4o, which are used to construct a small "cold-start" dataset. Since our reasoning pipeline begins from this stage, a natural question arises: to what extent does the final model depend on this proprietary model? We clarify that the primary role of GPT-4o in the cold-start phase is not to serve as a reasoning teacher but rather as a high-fidelity format generator. Its purpose is to produce a small seed dataset that precisely matches the structural requirements of our *Thinking-with-Image* reasoning format (e.g., correct use of `<think>` and `<tool_call>` tags). In other words, GPT-4o was utilized for its strong instruction-following capability, ensuring correct formatting rather than transferring reasoning ability. To examine the possible dependence on GPT-4o, we replaced it with two powerful open-source vision–language models, **Qwen3-VL-8B** and **InternVL3.5-8B**, for generating the same cold-start dataset. All subsequent stages—Reinforcement Fine-Tuning (RFT) and GRPO—were kept identical. The performance comparison across three evaluation benchmarks is summarized in Figure 6.

Across all cases, the cold-start stage alone shows small differences depending on the data source. However, these differences diminish rapidly after RFT and become negligible after GRPO. After the full training pipeline, all variants achieve nearly identical accuracy, thereby validating our conclusion that the framework is not dependent on any particular proprietary model.

## D PROMPTS TEMPLATES

In this section, we provide detailed prompt templates used across the workflow, including system prompts, input-pair construction templates, and templates or auxiliary prompts employed during synthetic data generation.

**System prompt** For our model, due to the presence of tool invocation, the following system prompt is used:

```
1 You are a helpful assistant.
2 Tools: You may call one or more functions to assist with the user query.
3 You are provided with function signatures within <tools></tools> XML
      tags:
```

```
1  <tools>:{
2      "type": "function",
3      "function": {
4          "name": "select_frames",
5          "description": "Select frames from a video.", "parameters": {
6              "type": "object",
7              "properties": {"target_frames": {
8              "type": "array",
9              "description": "List of frame indices to select from the
       video.",
10              "items": {"type": "integer", "description": "Frame index
       from 1 to N. N will be specified in the following"}}},
11          "required": ["target_frames"]}
12          }
13  }</tools>
14  For each function call, return a json object with function name and
       arguments within <tool_call></tool_call> XML tags:
15      <tool_call>
16          {"name": <function-name>, "arguments": <args-json-object>}
17      </tool_call>",
```

**Input data construction template**    Each input consists of a pair: a video preference datum and a query. The query is constructed following the prompt below. Notably, as discussed above, since the per-dimension annotations differ slightly across datasets, dataset-specific explanations are injected depending on the source of the video preference data.

```
1  Task Description:
2  Your task is to compare two videos generated based on the same prompt by
       analyzing their frames in detail and provide an overall judgment
       along with a judgment for each dimension. This involves:
3  - Iterative reasoning,
4  - Zooming in on details,
5  - Dynamically selecting frames for further analysis.
6
7  The provided frames are downsampled from these videos:
8  - Video 1: First four input frames.
9  - Video 2: Next four input frames.
10
11  The prompt is: {prompt}
12
13  Evaluation Dimensions:
14  1. {dim_name_1}(TA):
15      {dim_explain_1}
16  2. {dim_name_2}(VQ):
17      {dim_explain_2}
18  3. {dim_name_3}(MQ):
19      {dim_explain_3}
20
21  Frames and Analysis Rules
22  - 8 sampled frames are provided, evenly downsampled from {N} frames
23  - Insufficient frames? Request more:
24      <tool_call>{"target_frames": []}</tool_call>
25
26  Format Requirement:
27
28  1. Snapshot:
29  Every time you receive new visual information, summarize any information
       that might be useful for your final judgment within
       <Snapshot></Snapshot> tags.
30
31  2. Think:
```

```
1  Place all reasoning content within <Think></Think> tags.
2
3  3. **Answer**:
4  If the final answer can be determined, output the answer within
       <Answer></Answer> tags. If the answer is still uncertain, output the
       recommended answer and confidence level within <Recommend
       Answer></Recommend Answer> tags.
5  Here, 1 represents Video 1, 2 represents Video 2, and 0 represents Tie.
       The confidence levels range from high to low as 1, 2, and 3.
6
7  Examples:
8  <Answer>TA=1, VQ=1, MQ=0, OA=1</Answer>, or
9  <Recommend Answer>TA=0, VQ=1, MQ=0, OA=1, CF=2</Recommend Answer>
```

The per-dimension explanations for each dataset are as follows:

**VideoGen-Reward**

```
1  **Text Alignment**: Alignment between video content and prompt
2  **Visual Quality**: The visual aesthetics of the video
3  **Motion Quality**: Level of motion coherence
```

**Text2Video-Human**

```
1  **Alignment**: Alignment between video content and prompt
2  **Preference**: The intrinsic aesthetics of the video
3  **Coherence**: Level of temporal coherence
```

**MJ-Bench-Video**

```
1  **Alignment**: Alignment between video content and prompt
2  **Fineness**: The level of fineness in visual content
3  **Coherence & Consistency**: Level of temporal coherence and Consistency
```

**Cold Start Data Construction template**   To improve adherence to the response format and increase efficiency during the Cold Start data construction phase, we provide an additional example for Cold Start data generation (which is removed during training), as shown below:

```
1  **Reason Segment 1:**
2
3  **<Snapshot>**
4  The first four frames from Video 1 and Video 2 show a close-up of a
       mother orangutan holding her baby in the rainforest. The mother's
       expression is gentle, and the baby's tiny fingers are gripping her
       fur.
5
6  The background is lush with green foliage, and the lighting suggests it
       is raining. The frames capture the tender moment between the mother
       and her baby, aligning well with the caption.
7  **</Snapshot>**
8
9  **<think>**
10
11
12 The frames are clear and detailed, showing the intimate moment between
       the mother and her baby. The motion appears natural, and the visual
       quality is high, with vibrant colors and sharp details. The rain
       adds a serene and realistic touch to the scene. The frames are
       evenly sampled, ensuring a representative view of the video content.
```

```
1  The next four frames will help in evaluating the motion quality and
       visual quality more comprehensively. I will select frames 12, 24,
       36, 48, 60, 72, 84, and 96 to analyze further.
2  </think>
3
4
5  <recommend answer>
6  TA = 1, MQ = 0, VQ = 0, OA = 1, CF = 2
7  </recommend answer>
8
9  <tool_call>
10 {"name": "select_frames", "arguments": {"target_frames": [12, 24, 36,
       48, 60, 72, 84, 96]}}
11 </tool_call>
12
13 Reason Segment 2:
14
15 <Snapshot>
16 The first four frames from Video 1 and Video 2 show a close-up of a
       mother orangutan holding her baby in the rainforest. The mother's
       expression is gentle, and the baby's tiny fingers are gripping her
       fur.
17
18 The background is lush with green foliage, and the lighting suggests it
       is raining. The frames capture the tender moment between the mother
       and her baby, aligning well with the caption.
19
20 </Snapshot>
21
22 <think>
23 The selected frames provide a clear view of the motion quality and
       visual quality. The motion is smooth, with natural transitions and
       no abrupt changes.
24
25 The final frames confirm that Video 1 is superior in terms of motion
       quality, visual quality, and overall alignment with the caption. The
       confidence in this conclusion is high, as the selected frames
       provide clear evidence of the video's quality.
26 </think>
27
28 <final answer>
29 TA = 1, MQ = 1, VQ = 1, OA = 1
30 </final answer>
```

# E  FURTHER INSIGHTS AND DISCUSSION

## E.1  QUALITATIVE ANALYSIS OF FAILURE MODES

In this section, we present a qualitative analysis of the key reasoning failure modes observed in VIDEOSEARCH REASONER. Overall, VIDEOSEARCH REASONER performs remarkably well on video pairs that are similar to those in our training preference datasets—that is, pairs where both videos align semantically with the caption yet exhibit discernible differences in visual quality. However, when test cases deviate from this regime, a characteristic failure pattern could emerge, which we term **"overly cautious" reasoning**.

In this mode, the model engages in additional and often unnecessary rounds of frame selection and multimodal reasoning, even when the correct preference judgment is already apparent from the initial set of frames. Although this behavior rarely harms the final accuracy, it increases inference latency and introduces redundant tool invocations.

This failure mode most commonly arises in two scenarios:

**(i) One-sided caption alignment.** When one video perfectly follows the caption while the other is only partially related (e.g., the caption describes "a girl walking on the street," but the second video only shows the street itself), the model tends to suspect that the key difference may have been missed due to frame downsampling. Consequently, it performs additional zooming or frame retrieval steps to search for visual evidence—such as the presence of the girl—that simply does not exist.

**(ii) Near-identical video pairs.** When Video 1 and Video 2 are visually almost identical, the model sometimes hesitates to output the neutral judgment (`TA=0, VQ=0, MQ=0, OA=0`) directly. Instead, it conducts one or more redundant retrieval rounds in an attempt to detect potential key differences that it suspects might be hidden in frames omitted during downsampling which in fact do not exist.

The following reasoning chain serves as an illustrative case of the "overly cautious" mode. The caption is `''a girl is walking forward, camera moves down''`. In this example, `Video 1` aligns well with the caption, whereas `Video 2` depicts only a street scene with no person present. Nevertheless, the model performs an unnecessary additional retrieval round, attempting to locate the girl within `Video 2`.

```
 1  Reason Segment 1:
 2
 3  <Snapshot>
 4  The first four frames from Video 1 show a girl walking forward. The
        camera appears to move down, focusing on her legs and feet. The
        background is a park-like setting with greenery and sunlight. The
        frames are clear, and the girl's movement is smooth.
 5
 6  The first four frames from Video 2 depict a narrow street in a small
        town. The road appears empty, with no pedestrians visible. On either
        side of the path stand houses with classic European architectural
        features.
 7  </Snapshot>
 8
 9  <think>
10  The frames from Video 1 are consistent with the caption, showing the
        girl walking forward. The camera movement is smooth, and the visual
        quality is good. However, the walking girl does not appear in the
        frames from Video 2. Considering the information of video
        downsampling, it is possible that the girl's walking motion occurs
        in the frames that were skipped during sampling. Therefore, I need
        to zoom in on input frames 2-3, which correspond to frames 24-36 in
        the original video. I will select frames 24, 28, 32, 36, 84, 88, 92
        and 96 to analyze further.
11  </think>
12
13  <recommend answer>
14  TA = 1, MQ = 0, VQ = 1, OA = 1, CF = 3
15  </recommend answer>
16
17  <tool_call>
18  {"name": "select_frames", "arguments": {"target_frames": [24, 28, 32,
        36, 84, 88, 92, 96]}}
19  </tool_call>
20
21  Reason Segment 2:
22
23  <Snapshot>
24  The four frames from Video 1 provide a more detailed depiction of the
        girl walking forward. The scene is similar to the previous ones, but
        the girl's movement appears smoother and more refined. We can
        observe that the motion of her legs while walking is
        well-coordinated, and there are no signs of irregularity or
        stiffness in her body movements.
```

```
 1
 2  The four frames from Video 2 still show a narrow street in a small town.
        There are still no pedestrians visible, and the background elements,
        lighting conditions, and overall visual characteristics remain
        consistent with the earlier frames.
 3
 4  </Snapshot>
 5
 6  <think>
 7  The selected frames provide a clear view of both the motion quality and
        the visual content. The motion in Video 1 is smooth, with natural
        transitions and no abrupt changes. In terms of visual content, there
        is still no appearance of the walking girl in Video 2. Considering
        that additional frames have already been selected, we can be
        confident that Video 2 indeed lacks such motion.
 8
 9  The inclusion of more frames further confirms that Video 1 is superior
        in terms of motion quality, visual quality, and alignment with the
        caption. Given that the selected frames provide clear evidence of
        the video's content and overall quality, our confidence in this
        conclusion is strong.
10  </think>
11
12  <final answer>
13  TA = 1, MQ = 1, VQ = 1, OA = 1
14  </final answer>
```

We observe that such behaviors mainly appear in out-of-distribution or ambiguous scenarios. For typical preference data (caption-relevant video pairs with measurable quality differences), the model remains both accurate and reliable. Therefore, these failure cases do not actually affect the downstream applicability of our model, where it operates only on standard preference data. Nonetheless, a promising direction for future improvement is to augment training with a small, curated dataset specifically designed to teach the model to be more decisive in giving the final answer when the available visual evidence is already sufficient.

In essence, "overly cautious" reasoning reflects the model's inherent tendency to avoid premature conclusions. Targeted fine-tuning could effectively mitigate this behavior, enhancing both efficiency and robustness in future iterations of VIDEOSEARCH REASONER.

E.2  DISCUSSION ON TERMINOLOGY OF VISUAL REASONING OPERATION

In this section, we aim to clarify our use of the term "visual reasoning operation" and distinguish it from the more narrowly defined "visual retrieval operation." While the primary explicit mechanism in our framework is the select_frames tool, which indeed functions as a retrieval instrument, the core of our operation lies in the reasoning that precedes and dictates the retrieval action, not in the retrieval itself.

First, the invocation of the select_frames tool is not an isolated action but the culmination of a deliberate reasoning process. Before any tool is called, the model must analyze the available visual context, identify ambiguities or gaps in its understanding, and formulate a hypothesis about where to find decisive evidence. The decision of *whether* to seek more information and, more importantly, *which specific frames* to select, is a direct manifestation of the model's reasoning ability. The tool call is merely the mechanism for executing the plan formulated by this reasoning. Therefore, we conceptualize this as a holistic process where reasoning determines the necessity and parameters of a subsequent retrieval.

Second, in addition to the explicit tool call, our framework incorporates a vital implicit reasoning operation: the active summarization and compression of visual information. This occurs within our visual memory window mechanism, where visual content is distilled into the <snapshot> tag. As new frames are added, the model must actively integrate the new visual evidence with its existing understanding, synthesize the information, and distill the most salient points to carry forward in the

reasoning chain. This act of deciding what information is important enough to retain is a non-trivial cognitive task and represents a crucial form of visual reasoning.

Consequently, we contend that the terminology "visual reasoning operation" is both accurate and necessary. It clearly articulates the central role of reasoning throughout the entire process. This terminology emphasizes the model's **autonomous, deliberate cognitive engagement** with the visual content, which fundamentally differs from **passive, externally-driven mechanisms** implied by terms such as "visual retrieval" Our framework is not merely retrieving frames; it is actively reasoning about **what to retrieve**, **when to retrieve it**, and **how to integrate and compress** the retrieved information to advance toward a well-grounded quality assessment.

## F  LIMITATIONS

Our approach enhances the reward model through multimodal reasoning; however, this unavoidably introduces longer inference chains, leading to higher latency and computational cost . In future work, we will aim to reduce inference overhead and shorten Chain-of-Thought (CoT) length for straightforward video cases without compromising quality, by further improving the model's reasoning efficiency. Our current training pipeline primarily relies on Reject Fine-Tuning and GRPO, which tend to amplify capabilities the model has already learned (Yue et al., 2025). To achieve more substantial gains, constructing a higher-quality supervised fine-tuning dataset with carefully curated CoT rationales is essential. Building such datasets is an important direction for future research.

## G  USE OF LLMS

LLMs were used to support writing by suggesting improvements to grammar and clarity, and to assist with code authoring via completion. However, all final implementations and the experimental design were conceived and executed by the authors.

