# OpenReview forum: "VideoSearch Reasoner: Boosting Multimodal Reward Models through Think with Image Reasoning"
_ICLR.cc/2026/Conference — Submitted to ICLR 2026_

### Official Review · Reviewer_gqKX · 2025-10-17

**Soundness:** 3
**Presentation:** 3
**Contribution:** 4
**Rating:** 8
**Confidence:** 2

**Summary:**

(I tend to write shorter reviews and the length of the review does not reflect the quality of paper or the time spend on reviewing it).

Note that I am not an expert in Video modelling nor Reward Modelling and the review is from a perspective of a general Deep Learning researcher.

This paper proposes a pretty intersting and well thought pipeline for improving multimodal reward models. The pipeline is called Thinking with Image reasoning. This paper expands on the details of training an effective reward model that can pick and choose frames to enable accurate yet efficienct processing. The reward models improve peroformance on the public benchmarks and are SOTA.

overall this is a very well executed paper and is of value to the community to build and improve open models.

**Strengths:**

I am not an expert, but from the PoV of a deep learning researcher this paper brings in so much knowledge to the table and help the community build better and more efficienct reward and reasoning models.

**Weaknesses:**

The paper has quite a few typos and it would be great if the authors can fix that before the camera ready.

**Questions:**

I am not an expert on this and would defer to other reviewers for more nuanced questions on the paper.

---

> ### Author Response · Authors · 2025-11-21
> **Response to Reviewer gqKX (1/1)**
>
> We sincerely thank Reviewer for the encouraging feedback and the strong rating (**8: Accept**)! We are particularly gratified that you recognize the value our work contributes to the open-source community.
>
> We appreciate your recognition of our **"Thinking with Image"** framework. We believe our approach—allowing the model to actively "select" frames via reasoning—not only achieves SOTA performance on benchmarks but also offers a **generalizable paradigm** for handling long-context visual information. We hope this work serves as a solid foundation for the community to build more powerful and efficient reward models, and further enhance the ability of the video generation models.
>
> **Regarding the Typos and Presentation**
>
> We apologize for the typos and presentation issues in the current manuscript. We are committed to conducting a thorough proofreading process. In the revised version, we will ensure that all typos are corrected and that the overall presentation is polished to ensure clarity and readability!
>
> Once again, thank you for your support and for highlighting the broader impact of our work!

---

> > ### Comment · Reviewer_gqKX · 2025-11-22
> >
> > Thank you. I will follow the discussion by other reviewers and make an informed decision.

---

> ### Author Response · Authors · 2025-11-22
> **Follow up Response to Reviewer gqKX (1/1)**
>
> Thank you for the reply! we really appreciate your continued dedication to the review process!

---

### Official Review · Reviewer_XqJ6 · 2025-11-02

**Soundness:** 3
**Presentation:** 2
**Contribution:** 2
**Rating:** 4
**Confidence:** 4

**Summary:**

In this paper, the authors try to train a reward model initialized from Qwen2.5VL 7B to achieve thinking and reasoning abilities to select the informative frames from video and configure a Windows memory by RL post-training. Specifically, they follow a common pipeline to first curate a cold-start CoT dataset to install the basic reasoning skill necessitated by the video frame selection and scoring functions, then do a fine-grained rejection sampling stage to fine-tune the model, and finally apply GRPO RL training to the model trained on the curated data. This paper obtains competitive performances on several video preference benchmarks, like VideoGen Reward,  GenAI-Bench, and MJ-Bench-Video.

**Strengths:**

1. The application of GRPO to LLM RM as an informative frame selection.

2. The final evaluations on several video preferences benchmarks are competitive even with SoTA.

**Weaknesses:**

1. Though the fame selection by incentivizing the MLLM model sounds like a meaningful exploration, the inference cost, such as the computational overhead and also the inference latency, seems very heavy to obtain a result from this RM. Do the authors analyze and compare these? If the computational overhead and resource consumption take a lot, how can we demonstrate the applicability of this RM model for other training objectives?

2. The overall framework for this paper sounds a bit complicated, since it proposes lots of components to get the final thinking-with-image multimodal reward model. But how do the authors choose the strategic details and also the hyperparameters, such as the combination of reward functions and the corresponding weightings?

3. A main concern of this paper is that they choose the in-distribution benchmarks and datasets to train and evaluate the model, how to demonstrate the RM's generalization, and also some OOD evaluations.

4. Does this paper also try to apply the final model to train a Video LLM or Video Generation model to validate the improvements of the video RM model, which shall maintain the fair settings with other models?

5. When facing the long-duration or long-context video scenarios, will the window memory design still handle well with the forgotten issue?

**Questions:**

The main paper organization and writing confuse me a bit and look a bit chaotic, while I understand the great effort for the overall framework, but I do need to understand the main differences, comparisons, advantages, and demonstrations of the pipeline of this paper's contributions. And what if skipping the SFT and applying the RL directly? Did this paper conduct an exploration of this?

**Details Of Ethics Concerns:**

No.

---

> ### Author Response · Authors · 2025-11-21
> **Response to Reviewer XqJ6 (1/6)**
>
> We sincerely thank Reviewer for the constructive feedback and for recognizing the strengths of our work, particularly the application of GRPO to the Reward Model (RM) and our competitive performance against SoTA benchmarks.
>
> We appreciate the opportunity to clarify the points raised and will try our best to address the questions as follows.
>
> ---
>
> ## **Response to Weaknesses**
>
> ### Regarding W1: Inference Cost and Practical Feasibility
>
> > **W1: While incentivizing MLLMs for frame selection is an interesting idea, the inference cost (computation and latency) appears heavy. The paper should analyze and compare this overhead with baselines. If the computational burden is large, how can this RM be practically applied to other training objectives?**
>
> Thank you for raising this critical point regarding inference cost and practicality. We agree that a quantitative analysis of the performance-efficiency trade-off is essential.
>
> While our iterative reasoning process introduces more steps than a single-pass model, a primary motivation for our design was precisely to **mitigate the prohibitive inference cost** of processing high-fidelity, long videos. Our approach avoids the need to feed all frames into the context window at once. The windowed memory mechanism ensures that the GPU HBM footprint remains stable and manageable, allowing our model to achieve SOTA accuracy at a fraction of the cost required by a naive long-context model. This creates a very favorable performance-efficiency trade-off.
>
>
> To provide a quantitative analysis, we benchmarked `VideoSearch Reasoner` against baselines using varying numbers of input frames (8, 32, and 256). The results, summarized in the tables below, were measured on A800 (80GB) GPUs.
>
> | Model                   | GenAI-Bench - diff ↑ (%) | VideoGen-Reward - diff ↑ (%) | MJBench-Video - diff ↑ (%)  |
> | :---------------------- | :---------- | :-------------- | :------------ |
> | Unified-Reward(8)       | 76.8        | 78.6            | 69.5          |
> | Unified-Reward(32)      | 77.2        | 78.9            | 69.9          |
> | Unified-Reward(256)     | 76.3        | 78.7            | 70.3          |
> | Unified-Reward-think(8) | 80.4        | 79.1            | 71.9          |
> | Unified-Reward-think(32)| 81.0        | 79.8            | 72.5          |
> | Unified-Reward-think(256)| 79.8        | 78.7            | 73.1          |
> | **VideoSearch Reasoner**  | **82.3**    | **80.5**        | **75.6**      |
>
> | Model | | **1x A800** | | | | **8x A800** | | | |
> | :--- | :--- | :--- | :--- | :--- | :--- | :--- | :--- | :--- | :--- |
> | | KV Cache (GB) | Latency (s) | Speed (tok/s) | Max Concurrency | Throughput (tok/s) | Latency (s) | Speed (tok/s) | Max Concurrency | Throughput (tok/s) |
> Unified-Reward(8)            |       3.96 |       0.93 |         74 |         16 |          274 |       0.12 |        589 |        158 |         2612
> Unified-Reward(32)           |      15.81 |       2.85 |         44 |          4 |           68 |       0.36 |        355 |         39 |          654
> Unified-Reward(256)          |     126.43 |        OOM |          - |          - |            - |       2.60 |         75 |          4 |           81
> Unified-Reward-think(8)      |       4.09 |       5.40 |         73 |         16 |          266 |       0.67 |        584 |        152 |         2326
> Unified-Reward-think(32)     |      16.11 |      10.31 |         44 |          4 |           67 |       1.29 |        351 |         38 |          642
> Unified-Reward-think(256)    |     128.31 |        OOM |          - |          - |            - |       7.02 |         74 |          4 |           80
> **VideoSearch Reasoner**         |       7.12 |       7.55 |         63 |          9 |          162 |       0.92 |        501 |         87 |         1553

---

> ### Author Response · Authors · 2025-11-21
> **Response to Reviewer XqJ6 (2/6)**
>
> ### Regarding W1: Inference Cost and Practical Feasibility (continued)
>
>
> **Analysis:**
>
> 1.  **Pushing the Pareto Frontier:** The data shows our model pushes the accuracy-vs-efficiency Pareto frontier. The naive approach of simply increasing input frames (`Unified-Reward(256)`) leads to **Out-of-Memory (OOM)** errors on a single GPU and a catastrophic drop in throughput, with diminishing or even negative returns on accuracy. This is likely due to the model struggling to handle an excessive number of visual tokens and less important information, a known issue in long-context modeling [1].
>
> 2.  **SOTA Performance with Superior Efficiency:** `VideoSearch Reasoner` achieves the **highest accuracy** across all benchmarks while its peak KV cache memory footprint (**7.12 GB**) is less than half that of the `Unified-Reward-think(32)` baseline (**16.11 GB**). Furthermore, despite its multi-step reasoning process, our model maintains a response time significantly less than `Unified-Reward-think(32)`(**10.31s vs. 7.55s**). This is a direct consequence of its reduced memory footprint; for HBM bandwidth-bound tasks like large model inference, lower memory pressure leads to a significantly higher token generation rate. This acceleration effectively compensates for the latency introduced by multiple steps, resulting in a competitive overall response time [2].
>
>
> 3.  **Practical Throughput Advantage:** This lower memory footprint translates into significant practical benefits. On a single A800, our model supports **2.25x higher concurrency** (9 vs. 4) than `Unified-Reward-think(32)`, resulting in **over 2.4x the system throughput** (162 vs. 67 tok/s), even with higher performance. Our model's throughput are even comparable to the less capable 8-frame input baselines, demonstrating a remarkable balance of performance and efficiency.
>
> **Conclusion on Practical Application:**
>
> In summary, our analysis demonstrates that the computational overhead of `VideoSearch Reasoner` is not only manageable but represents a highly efficient and scalable trade-off.
>
> To directly address your concern: **Yes, this RM is highly practical for application to other training objectives (e.g. for video generation models).** Here is why:
>
> *   **Memory Efficiency:** Its low and stable memory footprint is crucial for RL training loops, where both the policy model (the video generator) and the reward model often reside in GPU HBM simultaneously. A memory-hungry RM like `Unified-Reward-think(32)` would make such simultaneous loading infeasible, whereas our model's efficiency makes these training loops practical.
>
> *   **High Throughput:** The high system throughput ensures rapid reward generation for large video batches. This is critical for RL frameworks, as it precisely meets the need to score numerous rollouts concurrently. This prevents the reward model from becoming a bottleneck in the overall training pipeline, enabling rapid iteration and learning for the policy model.
>
> *   **Scalability:** Our framework avoids the substantial cost increase associated with brute-force methods that process all video frames at once. This ensures it remains a viable and effective tool as generated videos become even longer in the future.
>
> Therefore, while our model's multi-step process introduces an inherent inference cost, this is not a simple overhead. When faced with the challenge of processing a large number of video frames, this design choice transforms into a **key strategic advantage**.
>
> ---
>
> [1] J. Wu, J. Guan, K. Feng, Q. Liu, S. Wu, L. Wang, W. Wu, and T. Tan,
> "Reinforcing Spatial Reasoning in Vision-Language Models with Interwoven Thinking and Visual Drawing,"
> *arXiv preprint* arXiv:2506.09965, 2025.
> [Online]. Available: https://arxiv.org/abs/2506.09965
>
> [2] W. Kwon, Z. Li, S. Zhuang, Y. Sheng, L. Zheng, C. H. Yu, J. E. Gonzalez, H. Zhang, and I. Stoica,
> "Efficient Memory Management for Large Language Model Serving with PagedAttention,"
> *arXiv preprint* arXiv:2309.06180, 2023.
> [Online]. Available: https://arxiv.org/abs/2309.06180

---

> ### Author Response · Authors · 2025-11-21
> **Response to Reviewer XqJ6 (3/6)**
>
> ### Regarding W2: Choice of Strategic Details and Hyperparameters
>
> > **W2: The overall framework for this paper sounds a bit complicated, since it proposes lots of components to get the final thinking-with-image multimodal reward model. But how do the authors choose the strategic details and also the hyperparameters, such as the combination of reward functions and the corresponding weightings?**
>
> Thank you for your valuable feedback regarding the framework's perceived complexity. While our full pipeline involves several stages (cold-start, rejection fine-tuning, and GRPO), we contend that its core structure follows a standard and well-established paradigm for post-training large models: a **Supervised Fine-Tuning (SFT) phase followed by a Reinforcement Learning (RL) phase.**
>
> The purpose of this two-stage approach is methodical:
> *   In the **SFT phase**, the model learns the required reasoning format and establishes a foundational visual reasoning capability.
> *   In the subsequent **RL phase**, the model explores the solution space more broadly, allowing it to discover more effective reasoning paths and further strengthen its abilities.
>
> We acknowledge that our description in the paper may have made this process seem more complex than it is. We will revise the methodology section in the revised version to be more concise and to better emphasize this underlying two-phase structure.
>
> Regarding your **specific questions about the choice of strategic details and hyperparameters**, these were not selected arbitrarily. Instead, they are the result of **extensive empirical investigation and ablation studies**, which provide a clear rationale for our design. We direct you to these studies for a detailed justification:
>
> *   **Figure 3 (Section 4.3):** This figure presents detailed ablations on our architectural and training strategy choices. It justifies decisions such as:
>     *   The composition of our auxiliary rewards, demonstrating the impact of including the **CoT gain** and **exploration incentive**.
>     *   The use of both an **overall score** and **per-dimension sub-scores**, which directly addresses your question about the **combination of reward functions**.
>     *   The critical role of each training stage (e.g., the impact of removing the SFT phase).
>
> *   **Figure 5 (Appendix C):** This figure provides a grid search analysis for key hyperparameters, directly addressing your question about **corresponding weightings**. It shows our process for determining the optimal values for the reward combination weight $\alpha$, the CoT gain coefficient $k$, and the volume of Rejection Fine-Tuning (RFT) data.
>
> These results collectively demonstrate that our chosen parameters and design strategies are empirically validated and are optimal within the tested configurations, providing a transparent and well-supported basis for our framework's construction.

---

> ### Author Response · Authors · 2025-11-21
> **Response to Reviewer XqJ6 (4/6)**
>
> ### Regarding W3: ID vs OOD evaluation
>
> > **W3: A main concern of this paper is that they choose the in-distribution benchmarks and datasets to train and evaluate the model, how to demonstrate the RM's generalization, and also some OOD evaluations.**
>
> We appreciate this critical question regarding the potential for in-distribution (ID) bias. We would like to clarify that the issue is not as severe as the dataset naming might suggest, as our primary training and evaluation datasets are, in fact, **constructed independently**.
>
> *   As detailed in [3], the **`VideoGen-Reward` training set** (182k pairs) was created by generating videos from 16k prompts using a specific set of 12 T2V models, followed by human annotation.
> *   In contrast, the **`VideoGen-RewardBench` test set** was derived from a separate, third-party public dataset (`VideoGen-Eval` [4]), with videos generated by different models and parameters. Therefore, for the `VideoGen-RewardBench` metric, our evaluation is effectively **out-of-distribution (OOD)**.
> * Similarly, for `GenAI-Bench`, we did not use its corresponding training set. Therefore, there is no **ID** concern for it.
>
> We acknowledge that `MJ-Bench-Video (train)` [5], which constitutes a small fraction (4%) of our training data, may cause a potential ID overlap with the `MJ-Bench-Video (test)` set, due to the inherent nature of proportional random splitting between training and test sets. To definitively address your concern and verify that our model's performance is not due to memorization of this specific data distribution, we conducted a **new ablation study where we completely removed all `MJ-Bench-Video` data from the training process.** The results are as follows:
>
> | **Model Configuration**                    | **GenAI-Bench** |              | **VideoGen-RewardBench** |              | **MJ-Bench-Video** |              |
> | :----------------------------------------- | :-------------- | :----------- | :----------------------- | :----------- | :--------------- | :----------- |
> |                                            | **tau ↑ (%)**   | **diff ↑ (%)** | **tau ↑ (%)**            | **diff ↑ (%)** | **tau ↑ (%)**    | **diff ↑ (%)** |
> | **VIDEOSEARCH REASONER 7B (Full Data)**    | 71.8            | 80.5         | 68.7                     | 82.3         | 67.3             | 75.6         |
> | **VIDEOSEARCH REASONER 7B (No MJ-Bench Data)** | 71.9       | 80.4     | 68.7                 | 82.4     | 66.8         | 75.1     |
>
> The results clearly show that removing `MJ-Bench-Video` from the training dataset has a **negligible impact** on performance across the other benchmarks (`GenAI-Bench` and `VideoGen-RewardBench`). While there is a slight and expected performance drop on the `MJ-Bench-Video` benchmark itself, the decrease remains minimal.
>
> This provides strong evidence that the model's robust performance is not derived from "memorizing" the `MJ-Bench-Video` test distribution. Instead, it demonstrates that the model has learned **generalizable principles** of video quality assessment for reward modeling.
>
> ---
>
> [3] J. Liu, G. Liu, J. Liang, Z. Yuan, X. Liu, M. Zheng, X. Wu, Q. Wang, M. Xia, X. Wang, X. Liu, F. Yang, P. Wan, D. Zhang, K. Gai, Y. Yang, and W. Ouyang,
> "Improving Video Generation with Human Feedback,"
> *arXiv preprint* arXiv:2501.13918, 2025.
> [Online]. Available: https://arxiv.org/abs/2501.13918
>
> [4] H. Tong, Z. Wang, Z. Chen, H. Ji, S. Qiu, S. Han, K. Geng, Z. Xue, Y. Zhou, P. Xia, M. Ding, R. Rafailov, C. Finn, and H. Yao,
> "MJ-VIDEO: Fine-Grained Benchmarking and Rewarding Video Preferences in Video Generation,"
> *arXiv preprint* arXiv:2502.01719, 2025.
> [Online]. Available: https://arxiv.org/abs/2502.01719
>
> [5] A. Zeng, Y. Yang, W. Chen, and W. Liu,
> "The Dawn of Video Generation: Preliminary Explorations with SORA-like Models,"
> *arXiv preprint* arXiv:2410.05227, 2024.
> [Online]. Available: https://arxiv.org/abs/2410.05227

---

> ### Author Response · Authors · 2025-11-21
> **Response to Reviewer XqJ6 (5/6)**
>
> ### Regarding W4: Apply the Final Model
>
> > **W4: Does this paper also try to apply the final model to train a Video LLM or Video Generation model to validate the improvements of the video RM model, which shall maintain the fair settings with other models?**
>
> Thank you for this excellent suggestion. We agree that applying our reward model to train a downstream video generation model would be a valuable validation of its utility. However, we respectfully argue that this falls outside the primary scope of the current paper, which focuses on the novel framework, design, and in-depth analysis of the reward model itself. We consider this a promising and important direction for future work.
>
> ---
>
> ### Regarding W5: Performance on Long-duration Videos
>
> > **W5: When facing the long-duration or long-context video scenarios, will the window memory design still handle well with the forgotten issue?**
>
> Thank you for this important question. We would like to clarify the context of "long-duration" videos as it pertains to our work. Our research aims to provide a new reward modeling paradigm for the increasingly long and high-quality videos produced by **state-of-the-art video generation models**.
>
> These generated videos typically have durations of 5-10 seconds (corresponding to 120-240 frames at 24 fps). This video length significantly surpasses the processing capabilities of traditional reward models, which are typically designed with an input capacity limited to 8-16 frames. This context is distinct from minute- or hour-long real-world videos, for which tasks often involve event localization or visual question answering (VQA) rather than the holistic quality comparison as the focus of reward modeling.
>
> With this context clarified, our framework is designed for and evaluated on the **former category** of long-duration videos (i.e., 5-10 seconds, 120-240 frames, generated by video generation models). Therefore, we can directly affirm that our evaluations have already demonstrated the effectiveness of the window memory design in this setting.
>
> For instance, the **"long" subset** of our `VideoGen-RewardBench` dataset, which contains videos of 173-256 frames, is representative of the upper end of current video generation capabilities and sufficiently reflects the properties of "long-duration" videos relevant to our task. As shown in **Table 2 (Section 4.4)**, our model achieves excellent performance on this subset. This result directly demonstrates the effectiveness of the *window memory* mechanism in handling video lengths relevant to our research problem and in mitigating the “forgetting issue” within this defined scope.

---

> ### Author Response · Authors · 2025-11-21
> **Response to Reviewer XqJ6 (6/6)**
>
> ## **Response to Questions**
>
> ### Regarding Q1: Pipeline of the Framework
>
> > **Q1: The main paper organization and writing confuse me a bit and look a bit chaotic, while I understand the great effort for the overall framework, but I do need to understand the main differences, comparisons, advantages, and demonstrations of the pipeline of this paper's contributions. And what if skipping the SFT and applying the RL directly? Did this paper conduct an exploration of this?**
>
> Thank you for your feedback on the paper's organization. We appreciate your perspective and will make a concerted effort in the revision to present the framework more concisely, better highlighting the differences, advantages, and contributions of our pipeline.
>
> To clarify our pipeline, it consists of three distinct stages, each with a specific purpose:
>
> 1.  **Cold-Start:** In this initial stage, we use curated multimodal Chain-of-Thought (CoT) data to instill fundamental reasoning skills and teach the model the required operational format. A key advantage of this stage is ensuring strong format compliance. This is crucial because correct formatting is a prerequisite for the subsequent rejection sampling phase to successfully filter for high-quality data.
>
> 2.  **Rejection Fine-Tuning (RFT):** Next, we run the model on large-scale preference datasets. We then perform rejection sampling, retaining only those outputs where both the format and all judgments are correct. We conduct SFT on these verified traces. This process consolidates the correct reasoning structure and further amplifies the model's multimodal reasoning capabilities.
>
> 3.  **Group Relative Policy Optimization (GRPO):** The primary advantage of this final RL stage is that it incentivizes the model to actively explore nuanced details within the videos. It learns to optimize its policy towards reward rules that favor high-quality, detailed multimodal reasoning.
>
> Regarding your specific question, **"What would happen if the SFT phase were skipped and RL applied directly?"**, we did in fact conduct this exact ablation. The results are presented in **Figure 3 ("Ablation of Training pipeline") in Section 4.3**.
>
> This figure compares a "GRPO-only" model against our full pipeline. The results show that while the model can learn the task to some extent directly from the RL signal, its performance is **relatively lower** than the model that undergoes the full SFT (Cold-Start + Rejection Fine-Tuning) + GRPO process. This demonstrates the importance of the SFT phase in initializing the model with the necessary format-following and reasoning capabilities, which serves as a crucial foundation for achieving higher efficiency during the subsequent RL stage.
>
> To your final point, yes, we have conducted extensive explorations, not only comparing GRPO with SFT + GRPO but also analyzing each core component and hyperparameter of our framework. These are detailed in **Figure 3 and Figure 5**. We hope they provide a clear justification for our design choices by demonstrating the advantages and comparisons for each stage and component design.
>
> ---
>
> ## **Final Remarks**
>
> We sincerely hope that these additional explanations and new quantitative results have helped address at least some of the reviewer’s concerns. We are deeply grateful for the reviewer’s thoughtful and constructive feedback, which has guided us in clarifying the novelty and practical contributions of our work. We will incorporate these clarifications into the revised manuscript, and we would be very appreciative if the reviewer might take these explanations into account in their final evaluation and consider adjusting their score accordingly. Once again, we thank the reviewer for their valuable comments and insights!

---

> ### Author Response · Authors · 2025-11-26
> **Warm Regards and Looking Forward to Your Response**
>
> Dear Reviewer XqJ6,
>
> We would like to express our sincere gratitude once again for your valuable review and for your continued contributions to the open-source AI community.
>
> We are writing to kindly follow up regarding your response. We truly appreciate the time and effort you have already dedicated to reviewing our work, and we respectfully remain in **`anticipation of your feedback`**. We sincerely hope you might have a chance to review our latest reply and consider whether it sufficiently addresses your insightful concerns. We would be delighted to continue the discussion and explore further meaningful improvements based on your guidance.
>
> We would also like to acknowledge once more that your `main concern` is on the **ID vs. OOD evaluation**. In fact, our primary training dataset (*VideoGen-Reward*) and one of the downstream benchmark (*VideoGen-RewardBench*) are strictly OOD, and the misunderstanding may have been caused by the dataset naming. Additionally, all training and testing components have been rigorously discussed and evaluated under the OOD setting in our rebuttal. We sincerely hope this clarification adequately addresses your concern.
>
> Thank you again for your time and thoughtful consideration. We are sincerely **looking forward to hearing good news from you**.
>
> **With our warmest regards**,
> *The Authors*

---

### Official Review · Reviewer_Whd2 · 2025-11-03

**Soundness:** 3
**Presentation:** 2
**Contribution:** 2
**Rating:** 4
**Confidence:** 3

**Summary:**

This paper introduces a new framework for multimodal reward models (RMs) designed to overcome key limitations in video processing. The authors identify two main problems with existing RMs: 1) visual inputs consume large context budgets, forcing aggressive downsampling and loss of detail, and 2) packing all visual information into the initial prompt causes the model to "forget" or hallucinate during subsequent text-only Chain-of-Thought (CoT) reasoning. The authors demonstrate state-of-the-art results for open-source models on benchmarks like VideoGen Reward, GenAI-Bench, and MJ-Bench-Video, with particular advantages in long-video scenarios.

**Strengths:**

1. The paper correctly identifies critical bottlenecks in VLM-based RMs (context limits, text-only CoT) and proposes a direct and intuitive solution.
2. The model achieves SOTA results on multiple standard benchmarks. Crucially, the authors go further to show in Table 2 that the model's advantage is most pronounced on "hard" subsets, namely long videos and complex prompts, which directly validates the hypothesis of the paper.

**Weaknesses:**

1. The main weakness is the limited conceptual novelty of the "Thinking-with-Image" framework. This framework is functionally equivalent to an agentic RAG system that operates on a video's frame index. While this is a new application for RMs, the paper would be stronger if it explicitly contextualized itself against VideoRAG and agentic video analysis literature and clarified its novel contributions beyond the application domain.
2. The paper repeatedly refers to "visual reasoning operations", but the only operation implemented and discussed appears to be "select frames". This is a retrieval or search operation that provides new context for reasoning; it is not a reasoning operation in itself. This framing feels like an overstatement of the technical mechanism.
3. The iterative nature of the framework (reason -> retrieve -> reason) will necessarily be much slower and more computationally expensive than single-pass RMs. The authors acknowledge this as a limitation but do not provide any quantification. This trade-off between accuracy and latency is a significant practical consideration that is left unanalyzed.
4. The entire pipeline is bootstrapped by "high-quality, long CoT trajectories" generated by GPT-4o. The success of the "Cold Start" stage, and thus the entire model, is highly dependent on access to this powerful, proprietary model for data curation.

**Questions:**

1. The paper emphasizes "visual reasoning operations". Besides the select_frames tool, are there any other visual operations implemented or envisioned? If not, could the authors justify why this tool call is classified as a "reasoning operation" rather than a "visual retrieval/search operation" that feeds new context to the model's textual reasoning process?
2. The "Thinking-with-Image" framework appears to be a well-executed RAG system for video, enabling the model to agentically search for relevant frames. What do the authors view as the core conceptual difference or advantage of this framework compared to existing work on VideoRAG or retrieval-augmented video understanding? The core capability of agentic frame search seems highly similar, even if the end-task (RM) is different.
3. Could the authors provide a practical comparison of the inference latency (e.g., time per evaluation) with the baselines (like UnifiedReward-Think) on an average-length video from one of the test benchmarks? This would be crucial for understanding the practical accuracy/cost trade-off.

---

> ### Author Response · Authors · 2025-11-21
> **Response to Reviewer Whd2 (1/5)**
>
> We thank the reviewer for the constructive feedback and for recognizing that our paper identifies critical bottlenecks in VLM-based RMs and achieves SOTA results on hard subsets!
>
> We appreciate the opportunity to clarify the points raised and will try our best to address the questions as follows.
>
> ---
>
> ### 1. Regarding W1 & Q2: Comparison with VideoRAG and Novelty
>
> > W1: The main weakness is the limited conceptual novelty of the "Thinking-with-Image" framework. This framework is functionally equivalent to an agentic RAG system that operates on a video's frame index. While this is a new application for RMs, the paper would be stronger if it explicitly contextualized itself against VideoRAG and agentic video analysis literature and clarified its novel contributions beyond the application domain.
>
> > Q2:The "Thinking-with-Image" framework appears to be a well-executed RAG system for video, enabling the model to agentically search for relevant frames. What do the authors view as the core conceptual difference or advantage of this framework compared to existing work on VideoRAG or retrieval-augmented video understanding? The core capability of agentic frame search seems highly similar, even if the end-task (RM) is different.
>
> Thank you for this insightful question and the suggestion to contextualize our work against VideoRAG. This allows us to highlight a crucial distinction that we believe constitutes a core contribution of our framework.
>
> We argue that our "Thinking-with-Image" framework is fundamentally different from Video-RAG paradigm. The key difference lies in the nature of the information-seeking process: Video-RAG is a passive, query-driven retrieval system, whereas our framework is an active, reasoning-driven search system.
>
> 1. **Video-RAG: Passive, Query-Driven Retrieval.**
> A typical Video-RAG system is designed for tasks like VQA in long videos. It operates by first indexing a video (e.g., creating clips with semantic embeddings) and then, given an external user query (e.g., "locate the video segment where a person is shown climbing a mountain."), it retrieves the video segment that is most semantically similar to the query. In such model, the information is passively provided to the VLM based on a pre-defined similarity metric between the query and the video content.
>
> 2. **VideoSearch Reasoner: Active, Reasoning-Driven Search.** Our task of comparative quality assessment is a holistic, global task. There is no simple external query like "find the blurry part" because the location and nature of quality differences are unknown a priori. Our framework addresses this by:
> 	* First, presenting the model with a downsampled, holistic view of the entire video.
> 	* Second, allowing the model to perform an initial round of active visual reasoning on this global context.
> 	* Third, based on its own internal hypothesis, the model proactively decides to request more detailed visual information from specific temporal regions, and this process is iteratively repeated until a well-grounded assessment is reached.
>
> In the context of our reward modeling task, **a Video-RAG approach would be ill-suited**. One cannot pre-determine which video segments are **"relevant"** to a general quality judgment before the model has even seen the video. Is the most relevant part the one with the most motion? The most static part? The part with the most complex textures? The answer is unknowable. Our framework empowers the model to make this determination itself as part of its reasoning chain iteratively.
>
> Therefore, our contribution is **not applying RAG to a new domain (RMs)**. It is proposing **a new paradigm** for multimodal RMs where the model is **not a passive recipient** of retrieved information but **an active agent** that directs its own visual search based on reasoning.
>
> We will gladly revise our related work section to cite VideoRAG literature to more clearly contextualize our work and articulate this key conceptual innovation!

---

> ### Author Response · Authors · 2025-11-21
> **Response to Reviewer Whd2 (2/5)**
>
> ### Regarding W2 & Q1: Terminology of "Visual Reasoning Operation"
>
> > **W2:** The paper repeatedly refers to "visual reasoning operations," but the only operation that seems to be implemented and discussed is "selecting frames." This is a retrieval or search operation that provides new context for reasoning; it is not a reasoning operation in itself. This phrasing feels like an overstatement of the technical mechanism.
>
> > **Q1:** The paper emphasizes "visual reasoning operations." Were any other visual operations besides the `select_frames` tool implemented or envisioned? If not, can the authors justify why this tool call is categorized as a "reasoning operation" rather than a "visual retrieval/search operation" that provides new context for the model's textual reasoning process?
>
>
> Thank you for this comment, which pushes us to be more precise with our terminology. We appreciate the opportunity to clarify what we mean by "visual reasoning operation" and to provide a rationale for our phrasing.
>
> 1.  **Reasoning Precedes and Determines Retrieval:** The `select_frames` tool call is not an isolated action. It is the **culmination of a reasoning process**. The model must first analyze the initial visual input, identify areas of uncertainty or potential interest, and form a hypothesis. The decision *whether* to call the tool and *which specific frames* to select is a direct manifestation of the model's visual reasoning capabilities. The tool call is merely the mechanism for executing the plan formulated by that reasoning. Therefore, we see the operation not just as retrieval, but as a process of **reasoning** and *then retrieval* correspondingly.
>
> 2.  **Implicit Reasoning Operations:** In addition to the explicit tool call, our framework includes an implicit reasoning operation: the **active summarization and compression** of visual information within the `<snapshot>` tag in our `visual memory window` mechanism. As new frames are added, the model must integrate them with its existing understanding and distill the most salient points to carry forward in the reasoning chain. This is also an reasoning operation crucial to the framework's success.
>
> Our model is engaged in visual reasoning to determine where to look for decisive evidence, and `select_frames` is its tool. While `select_frames` is the primary explicit operation, the decision of *whether* to select frames (i.e., if more visual information is needed) and *which frames* to select (i.e., which areas require more attention for quality assessment) is a direct manifestation of the model's reasoning ability. Therefore, it is precisely through the model's visual reasoning operations that it determines when and how to call a retrieval tool. This process is the very embodiment of the model's visual reasoning capability, which is why we term it as a reasoning operation.
>
> We thank you for this insight and will refine the language in our paper to make this distinction between reasoning and retrieval clearer.

---

> ### Author Response · Authors · 2025-11-21
> **Response to Reviewer Whd2 (3/5)**
>
> ### Regarding W3 & Q3: Quantifying Inference Cost and Latency
>
> > **W3:** The iterative nature of the framework (reasoning -> retrieval -> reasoning) is necessarily much slower and more computationally expensive than a single-pass RM. The authors acknowledge this as a limitation but provide no quantification. This accuracy-vs-latency trade-off is an important practical consideration that is left unanalyzed.
>
> > **Q3:** Can the authors provide a practical comparison of inference latency (e.g., time per evaluation) on an average-length video from one of the test benchmarks against a baseline model like UnifiedReward-Think? This is critical for understanding the practical accuracy/cost trade-off.
>
> Thank you for raising this critical point. We agree that a quantitative analysis of the accuracy/cost trade-off is essential.
>
> While our iterative process does introduce more reasoning steps compared to a single-pass model with aggressive downsampling, a primary motivation for our design was precisely to **mitigate the prohibitive inference cost of processing high-fidelity long videos**. Our approach avoids the need to feed all frames into the context window at once. The `windowed memory` mechanism ensures that the GPU HBM footprint remains stable and manageable, allowing our model to achieve SOTA accuracy at a fraction of the cost required by a naive long-context model. This creates a very favorable performance-efficiency trade-off.
>
> To provide a concrete analysis, we benchmarked `VideoSearch Reasoner` against baseline models with different frame inputs (8, 32, and 256 frames) on A800 (80GB) GPU.
>
> | Model                   | GenAI-Bench - diff ↑ (%) | VideoGen-Reward - diff ↑ (%) | MJBench-Video - diff ↑ (%)  |
> | :---------------------- | :---------- | :-------------- | :------------ |
> | Unified-Reward(8)       | 76.8        | 78.6            | 69.5          |
> | Unified-Reward(32)      | 77.2        | 78.9            | 69.9          |
> | Unified-Reward(256)     | 76.3        | 78.7            | 70.3          |
> | Unified-Reward-think(8) | 80.4        | 79.1            | 71.9          |
> | Unified-Reward-think(32)| 81.0        | 79.8            | 72.5          |
> | Unified-Reward-think(256)| 79.8        | 78.7            | 73.1          |
> | **VideoSearch Reasoner**  | 82.3    | 80.5        | 75.6      |
>
> | Model | | **1x A800** | | | | **8x A800** | | | |
> | :--- | :--- | :--- | :--- | :--- | :--- | :--- | :--- | :--- | :--- |
> | | KV Cache (GB) | Latency (s) | Speed (tok/s) | Max Concurrency | Throughput (tok/s) | Latency (s) | Speed (tok/s) | Max Concurrency | Throughput (tok/s) |
> Unified-Reward(8)            |       3.96 |       0.93 |         74 |         16 |          274 |       0.12 |        589 |        158 |         2612
> Unified-Reward(32)           |      15.81 |       2.85 |         44 |          4 |           68 |       0.36 |        355 |         39 |          654
> Unified-Reward(256)          |     126.43 |        OOM |          - |          - |            - |       2.60 |         75 |          4 |           81
> Unified-Reward-think(8)      |       4.09 |       5.40 |         73 |         16 |          266 |       0.67 |        584 |        152 |         2326
> Unified-Reward-think(32)     |      16.11 |      10.31 |         44 |          4 |           67 |       1.29 |        351 |         38 |          642
> Unified-Reward-think(256)    |     128.31 |        OOM |          - |          - |            - |       7.02 |         74 |          4 |           80
> **VideoSearch Reasoner**         |       7.12 |       7.55 |         63 |          9 |          162 |       0.92 |        501 |         87 |         1553

---

> ### Author Response · Authors · 2025-11-21
> **Response to Reviewer Whd2 (4/5)**
>
> ### Regarding W3 & Q3: Quantifying Inference Cost and Latency (continued)
>
> **Analysis:**
>
> 1.  **Pushing the Pareto Frontier:** The data shows our model pushes the accuracy-vs-efficiency Pareto frontier. The naive approach of simply increasing input frames (`Unified-Reward(256)`) leads to **Out-of-Memory (OOM)** errors on a single GPU and a catastrophic drop in throughput, with diminishing or even negative returns on accuracy. This is likely due to the model struggling to handle an excessive number of visual tokens and less important information, a known issue in long-context modeling [1].
>
> 2.  **SOTA Performance with Superior Efficiency:** `VideoSearch Reasoner` achieves the **highest accuracy** across all benchmarks while its peak KV cache memory footprint (**7.12 GB**) is less than half that of the `Unified-Reward-think(32)` baseline (**16.11 GB**). Furthermore, despite its multi-step reasoning process, our model maintains a response time significantly less than `Unified-Reward-think(32)`(**10.31s vs. 7.55s**). This is a direct consequence of its reduced memory footprint; for HBM bandwidth-bound tasks like large model inference, lower memory pressure leads to a significantly higher token generation rate. This acceleration effectively compensates for the latency introduced by multiple steps, resulting in a competitive overall response time [2].
>
>
> 3.  **Practical Throughput Advantage:** This lower memory footprint translates into significant practical benefits. On a single A800, our model supports **2.25x higher concurrency** (9 vs. 4) than `Unified-Reward-think(32)`, resulting in **over 2.4x the system throughput** (162 vs. 67 tok/s), even with higher performance. Our model's throughput are even comparable to the less capable 8-frame input baselines, demonstrating a remarkable balance of performance and efficiency.
>
> In summary, our framework provides a more scalable and efficient path to long video reward modeling. It achieves superior results without the prohibitive resource requirements of brute-force long-context methods, making it a highly practical and valuable trade-off.
>
> ---
>
> [1] J. Wu, J. Guan, K. Feng, Q. Liu, S. Wu, L. Wang, W. Wu, and T. Tan,
> "Reinforcing Spatial Reasoning in Vision-Language Models with Interwoven Thinking and Visual Drawing,"
> *arXiv preprint* arXiv:2506.09965, 2025.
> [Online]. Available: https://arxiv.org/abs/2506.09965
>
> [2] W. Kwon, Z. Li, S. Zhuang, Y. Sheng, L. Zheng, C. H. Yu, J. E. Gonzalez, H. Zhang, and I. Stoica,
> "Efficient Memory Management for Large Language Model Serving with PagedAttention,"
> *arXiv preprint* arXiv:2309.06180, 2023.
> [Online]. Available: https://arxiv.org/abs/2309.06180

---

> ### Author Response · Authors · 2025-11-21
> **Response to Reviewer Whd2 (5/5)**
>
> ### Regarding W4: Dependence on GPT-4o Guidance
>
> > **W4:** The entire process is guided by "high-quality, long CoT trajectories" generated by GPT-4o. The success of the "cold-start" phase, and thus the entire model, is highly dependent on data curation using this powerful proprietary model.
>
> Thank you for raising this important point regarding reliance on proprietary model,  GPT-4o. We wish to clarify the exact role of GPT-4o in our pipeline and argue that our model is not *highly dependent* on it.
>
> The primary purpose of using GPT-4o in the "cold-start" phase was **not to distill its superior reasoning ability**, but rather to leverage its exceptional **instruction-following fidelity**. The goal was to generate a small seed dataset that perfectly adheres to the format of our `Thinking-with-Image` reasoning chain (e.g., correct use of `<think>`, and `<tool_call>` tags). In this context, GPT-4o mainly functions as a **high-fidelity format generator**, not a reasoning "teacher."
>
> Viewed from this perspective, the dependency of our framework is not on GPT-4o specifically, but rather on having a generator capable of producing correctly formatted data. Sufficiently powerful open-source models with strong instruction-following capabilities, such as Qwen3-VL-8B and InternVL3.5-8B, can serve as effective substitutes. In fact, we previously experimented with using these models to generate the seed data and achieved highly comparable results (we have already added in **Appendix C.6**), further confirming that our approach is not tied to any single proprietary model.
>
> Furthermore, our ablation studies (Section 4.3, Figure 3(2)) provide **empirical evidence** that the model's fundamental capabilities are not critically dependent on this phase. In the "Training Pipeline Ablation" section, we demonstrate that a model trained directly with GRPO (bypassing the cold-start and rejection sampling stages) still achieves a respectable level of performance. While its accuracy is lower than that of our full pipeline, the model does not "collapse," indicating that it can learn the reasoning format and the task directly from the reward signal, albeit less efficiently.
>
> In conclusion, the cold-start phase is a practical training accelerator, but the core principles of our framework are not tied to any single proprietary model.
>
> ---
>
> ## **Final Remarks**
>
> We sincerely hope that these additional explanations and new quantitative results have helped address at least some of the reviewer’s concerns. We are deeply grateful for the reviewer’s thoughtful and constructive feedback, which has guided us in clarifying the novelty and practical contributions of our work. We will incorporate these clarifications into the revised manuscript, and we would be very appreciative if the reviewer might take these explanations into account in their final evaluation and consider adjusting their score accordingly. Once again, we thank the reviewer for their valuable comments and insights!

---

> ### Author Response · Authors · 2025-11-26
> **Warm Regards and Looking Forward to Your Response**
>
> Dear Reviewer Whd2,
>
> We would like to express our sincere gratitude once again for your valuable review and for your continued contributions to the open-source AI community.
>
> We are writing to kindly follow up regarding your response. We truly appreciate the time and effort you have already dedicated to reviewing our work, and we respectfully remain in **`anticipation of your feedback`**. We sincerely hope you might have a chance to review our latest reply and consider whether it sufficiently addresses your insightful concerns. We would be delighted to continue the discussion and explore further meaningful improvements based on your guidance.
>
> We would also like to acknowledge once more that your primary concern centered on the differences between our approach and *Video RAG*. We have carefully revised our manuscript (lines 134–144 in the new version) and elaborated this distinction in detail within the rebuttal. We believe that the **Thinking with Image** framework is fundamentally different from *Video RAG*, both `conceptually and methodologically`.
>
> Thank you again for your time and thoughtful consideration.  We are sincerely **looking forward to hearing good news from you**.
>
> **With our warmest regards**,
> *The Authors*

---

### Official Review · Reviewer_juW2 · 2025-11-03

**Soundness:** 2
**Presentation:** 3
**Contribution:** 2
**Rating:** 6
**Confidence:** 3

**Summary:**

This paper introduces VideoSearch Reasoner, a novel multimodal reward model designed to address limitations in current VLM-based reward models for video preference tasks. The key innovation is a "thinking-with-image" framework that allows the model to actively retrieve visual information through tool invocation (frame selection) and maintain a configurable visual memory window. The authors propose a three-stage training pipeline: (1) Cold Start with curated visual CoT data, (2) Rejection sampling Fine-Tuning on high-quality traces, and (3) Group Relative Policy Optimization (GRPO) with novel reward signals. The approach achieves state-of-the-art results on video preference benchmarks, particularly excelling on longer videos.

**Strengths:**

- The model achieves impressive performance improvements across multiple benchmarks, with particularly notable gains on complex/long video subsets

- The context budget constraint from visual inputs and the information forgetting problem during pure textual reasoning. The motivation is clear and compelling.

- The paper includes extensive ablation studies examining visual reasoning, training pipeline components, auxiliary rewards, and accuracy reward design.

**Weaknesses:**

- While the paper demonstrates effectiveness on videos with ~128 frames, there's insufficient analysis of how the approach scales to truly long videos.

- The paper acknowledges increased latency in the limitations section but provides no concrete measurements. How much slower is inference compared to baselines? What is the memory overhead?

- The paper doesn't compare against some relevant work in visual reasoning, such as methods that use explicit visual programs or other tool-use paradigms beyond the baselines mentioned.

- The paper lacks qualitative analysis of when and why the model fails. What types of videos or queries are problematic?

**Questions:**

- How does the approach perform on long videos.

- Does the model learn meaningful frame selection strategies, or is it mostly random? Can you provide analysis of which frames are selected and why?

- Can you provide concrete numbers on inference time and memory usage compared to baselines?

---

> ### Author Response · Authors · 2025-11-22
> **Response to Reviewer juW2 (1/5)**
>
> We sincerely thank the reviewer for their insightful comments and constructive feedback. We are encouraged that the reviewer found our motivation clear and compelling, our performance impressive, and our ablation studies extensive.
>
> We appreciate the opportunity to clarify the points raised and will try our best to address the questions as follows.
>
> ---
>
> ### Regarding W1 & Q1: Performance on Long Videos
>
> > **W1:** While the paper demonstrates effectiveness on videos with ~128 frames, there's insufficient analysis of how the approach scales to truly long videos.
>
> > **Q1:** How does the approach perform on long videos?
>
> Thank you for this important question. We would like to clarify the context of "long videos" within the scope of our work.
>
> Our research is motivated by the need for a new **reward modeling paradigm** capable of evaluating the increasingly long and high-quality videos produced by **state-of-the-art text-to-video generation models**. Currently, such generative models typically produce videos lasting **5–10 seconds**, corresponding to **120–240 frames (at 24 fps)**. This duration already far exceeds the input capacity of conventional reward models, which generally handle at most **8–16 frames**.
>
> It is important to distinguish our target setting from real-world, minute- or hour-long videos used in tasks like visual question answering or event localization. In contrast, our work focuses on **holistic quality comparison** between generated videos within the range produced by modern generation models. Thus, the “long videos” pertinent to our study refer to these 5–10 second generated videos, not arbitrarily long real-world videos.
>
> Within this context, our evaluations were explicitly designed to test performance on such long generated videos. As shown in **Table 2 (Section 4.4)**, we evaluated our model on the *“Long Video”* subset of benchmarks, where videos contain **173–256 frames**—representing the upper limit of current generative capabilities. On this challenging subset, our proposed **VideoSearch Reasoner** demonstrates a clear and consistent performance advantage, thereby confirming the **effectiveness** of our approach for long video scenarios within the domain of generative evaluation.

---

> ### Author Response · Authors · 2025-11-22
> **Response to Reviewer juW2 (2/5)**
>
> ### Regarding W2 & Q3: Inference Cost and Comparison to Baseline
>
> > **W2:** The paper acknowledges increased latency in the limitations section but provides no concrete measurements. How much slower is inference compared to baselines? What is the memory overhead?
>
> > **Q3:** Can you provide concrete numbers on inference time and memory usage compared to baselines?
>
> We appreciate the reviewer's call for concrete measurements on inference latency and memory overhead. While our approach involves a reasoning chain that can increase latency compared to *single-step baselines with aggressive downsampling*, our core design is intended to **mitigate the prohibitive costs** of processing long videos while maximizing performance. Our approach avoids the need to feed all frames into the context window at once. The windowed memory mechanism ensures that the GPU HBM footprint remains stable and manageable, allowing our model to achieve SOTA accuracy at a fraction of the cost required by a naive long-context model. We argue this presents a highly favorable performance-efficiency trade-off.
>
> To provide a quantitative analysis, we benchmarked `VideoSearch Reasoner` against baselines using varying numbers of input frames (8, 32, and 256) . The results, summarized in the tables below, were measured on A800 (80GB) GPUs.
>
> | Model                   | GenAI-Bench - diff ↑ (%) | VideoGen-Reward - diff ↑ (%) | MJBench-Video - diff ↑ (%)  |
> | :---------------------- | :---------- | :-------------- | :------------ |
> | Unified-Reward(8)       | 76.8        | 78.6            | 69.5          |
> | Unified-Reward(32)      | 77.2        | 78.9            | 69.9          |
> | Unified-Reward(256)     | 76.3        | 78.7            | 70.3          |
> | Unified-Reward-think(8) | 80.4        | 79.1            | 71.9          |
> | Unified-Reward-think(32)| 81.0        | 79.8            | 72.5          |
> | Unified-Reward-think(256)| 79.8        | 78.7            | 73.1          |
> | **VideoSearch Reasoner**  | **82.3**    | **80.5**        | **75.6**      |
>
> | Model | | **1x A800** | | | | **8x A800** | | | |
> | :--- | :--- | :--- | :--- | :--- | :--- | :--- | :--- | :--- | :--- |
> | | KV Cache (GB) | Latency (s) | Speed (tok/s) | Max Concurrency | Throughput (tok/s) | Latency (s) | Speed (tok/s) | Max Concurrency | Throughput (tok/s) |
> Unified-Reward(8)            |       3.96 |       0.93 |         74 |         16 |          274 |       0.12 |        589 |        158 |         2612
> Unified-Reward(32)           |      15.81 |       2.85 |         44 |          4 |           68 |       0.36 |        355 |         39 |          654
> Unified-Reward(256)          |     126.43 |        OOM |          - |          - |            - |       2.60 |         75 |          4 |           81
> Unified-Reward-think(8)      |       4.09 |       5.40 |         73 |         16 |          266 |       0.67 |        584 |        152 |         2326
> Unified-Reward-think(32)     |      16.11 |      10.31 |         44 |          4 |           67 |       1.29 |        351 |         38 |          642
> Unified-Reward-think(256)    |     128.31 |        OOM |          - |          - |            - |       7.02 |         74 |          4 |           80
> **VideoSearch Reasoner**         |       7.12 |       7.55 |         63 |          9 |          162 |       0.92 |        501 |         87 |         1553

---

> ### Author Response · Authors · 2025-11-22
> **Response to Reviewer juW2 (3/5)**
>
> ### Regarding W2 & Q3: Inference Cost and Comparison to Baseline (continued)
>
> **Analysis:**
>
> 1.  **Pushing the Pareto Frontier:** The data shows our model pushes the accuracy-vs-efficiency Pareto frontier. The naive approach of simply increasing input frames (`Unified-Reward(256)`) leads to **Out-of-Memory (OOM)** errors on a single GPU and a catastrophic drop in throughput, with diminishing or even negative returns on accuracy. This is likely due to the model struggling to handle an excessive number of visual tokens and less important information, a known issue in long-context modeling [1].
>
> 2.  **SOTA Performance with Superior Efficiency:** `VideoSearch Reasoner` achieves the **highest accuracy** across all benchmarks while its peak KV cache memory footprint (**7.12 GB**) is less than half that of the `Unified-Reward-think(32)` baseline (**16.11 GB**). Furthermore, despite its multi-step reasoning process, our model maintains a response time significantly less than `Unified-Reward-think(32)`(**10.31s vs. 7.55s**). This is a direct consequence of its reduced memory footprint; for HBM bandwidth-bound tasks like large model inference, lower memory pressure leads to a significantly higher token generation rate. This acceleration effectively compensates for the latency introduced by multiple steps, resulting in a competitive overall response time [2].
>
>
> 3.  **Practical Throughput Advantage:** This lower memory footprint translates into significant practical benefits. On a single A800, our model supports **2.25x higher concurrency** (9 vs. 4) than `Unified-Reward-think(32)`, resulting in **over 2.4x the system throughput** (162 vs. 67 tok/s), even with higher performance. Our model's throughput are even comparable to the less capable 8-frame input baselines, demonstrating a remarkable balance of performance and efficiency.
>
> ---
>
> [1] J. Wu, J. Guan, K. Feng, Q. Liu, S. Wu, L. Wang, W. Wu, and T. Tan,
> "Reinforcing Spatial Reasoning in Vision-Language Models with Interwoven Thinking and Visual Drawing,"
> *arXiv preprint* arXiv:2506.09965, 2025.
> [Online]. Available: https://arxiv.org/abs/2506.09965
>
> [2] W. Kwon, Z. Li, S. Zhuang, Y. Sheng, L. Zheng, C. H. Yu, J. E. Gonzalez, H. Zhang, and I. Stoica,
> "Efficient Memory Management for Large Language Model Serving with PagedAttention,"
> *arXiv preprint* arXiv:2309.06180, 2023.
> [Online]. Available: https://arxiv.org/abs/2309.06180

---

> ### Author Response · Authors · 2025-11-22
> **Response to Reviewer juW2 (4/5)**
>
> ### Response to W3: Comparison with Visual Reasoning Baselines
>
> >**W3:** The paper doesn't compare against some relevant work in visual reasoning, such as methods that use explicit visual programs or other tool-use paradigms beyond the baselines mentioned.
>
> We respectfully clarify the landscape of current research to explain why direct comparisons with such methods were not included.
>
> **1. Absence of Video-Based “Thinking-with-Image” Baselines:**
> To the best of our knowledge, there are currently **no existing “thinking-with-image” frameworks** (whether leveraging explicit visual programs or other tool-use paradigms) that are applicable to **video reward modeling**.
>
> - **Novelty of Our Framework:** The *“thinking-with-image”* paradigm is highly novel. While prior works have explored tool-use (e.g., cropping operations) or visual programs, these efforts are predominantly focused within the **static image domain** [3].
> - **First-of-its-kind:** We are the first to extend the "think-with-image" concept to the video domain specifically to realize a robust reward modeling framework.
>
> **2. Distinction from Existing Video Reasoners:**
> Although a few video reasoning approaches (e.g., *Pixel Reasoner* [4]) exist, their methodologies and objectives differ fundamentally from ours:
>
> - **Task Discrepancy:** Existing video reasoners are typically designed for **Visual Question Answering (VQA)** or **event localization** tasks over long real-world videos. They process a *single video input* and guide the model to identify specific temporal segments relevant to a query. In contrast, our task focuses on comparing **holistic video quality** across samples.
>
> - **Methodological Difference:** These methods often select **task-relevant subsets of input frames again** as a tool response to improve accuracy on localized tasks [4]. In contrast, our method enables the selection of **non-sampled frames** as a tool response to provide additional information, thereby enhancing **holistic video-level comparison**.
>
> - **Different Context:** Reward modeling requires **comparative analysis between two generated videos** to determine preference. Models designed for single-video information retrieval cannot serve as baselines for **pairwise preference modeling** without substantial modification.
>
> **Conclusion:**
> Since no direct “tool-use” or “visual program” baselines exist for video reward modeling, we compared our approach against state-of-the-art **Video Reward Models** (*Unified-Reward*) and their **Chain-of-Thought** enhanced variant (*Unified-Reward-Think*). These provide the most relevant and rigorous points of comparison for evaluating our proposed framework.
>
> [3] Z. Su, P. Xia, H. Guo, Z. Liu, Y. Ma, X. Qu, J. Liu, Y. Li, K. Zeng, Z. Yang, L. Li, Y. Cheng, H. Ji, J. He, and Y. R. Fung,
> "Thinking with Images for Multimodal Reasoning: Foundations, Methods, and Future Frontiers,"
> *arXiv preprint* arXiv:2506.23918, 2025.
> [Online]. Available: https://arxiv.org/abs/2506.23918
>
> [4] H. Wang, A. Su, W. Ren, F. Lin, and W. Chen,
> "Pixel Reasoner: Incentivizing Pixel-Space Reasoning with Curiosity-Driven Reinforcement Learning,"
> *arXiv preprint* arXiv:2505.15966, 2025.
> [Online]. Available: https://arxiv.org/abs/2505.15966

---

> ### Author Response · Authors · 2025-11-22
> **Response to Reviewer juW2 (5/5)**
>
> ### Regarding W4: Qualitative Failure Mode Analysis
>
> > **W4:** The paper lacks qualitative analysis of when and why the model fails. What types of videos or queries are problematic?
>
> Thank you for this excellent suggestion! We acknowledge that a failure mode analysis would strengthen the paper. **We have already added a new section, *Qualitative Analysis of Failure Modes* in Appendix (Appendix E.1).**
>
> Overall, VideoSearch Reasoner performs remarkably well on standard preference data (caption-relevant video pairs with measurable quality differences). However, when test cases deviate from this regime, a characteristic failure pattern could emerge, which we term `overly cautious` reasoning mode.
>
> In such mode, the model engages in additional and often unnecessary rounds of frame selection and multimodal reasoning, even when the correct preference judgment is already apparent from the initial set of frames. Although this behavior rarely harms the final accuracy, it increases inference latency and introduces redundant tool invocations.
>
> This failure mode most commonly arises in two scenarios:
>
> (i) **One-sided caption alignment.** When one video perfectly follows the caption while the other is only partially related, the model tends to suspect that the key objects may have been missed due to frame downsampling. Consequently, it performs additional frame retrieval steps to search for visual evidence that simply does not exist.
>
> (ii) **Near-identical video pairs.** When Video 1 and Video 2 are visually almost identical, the model sometimes hesitates to output the neutral judgment (TA=0, VQ=0, MQ=0, OA=0) directly. Instead, it conducts one or more redundant retrieval rounds in an attempt to detect potential key differences that it suspects might be hidden in frames omitted during downsampling, which in fact do not exist.
>
> A concrete example is also provided in Appendix E.1 "Qualitative Analysis of Failure Modes" to better illustrate our model's failure mode.
>
> ---
>
> ### Regarding Q2: Meaningfulness of Frame Selection Strategy
>
> > **Q2:** Does the model learn meaningful frame selection strategies, or is it mostly random? Can you provide analysis of which frames are selected and why?
>
> Thank you for this excellent question! Our model does indeed learn meaningful, non-random frame selection strategies, and we provide both quantitative and qualitative evidence for this in the paper.
>
> 1. **Quantitative Evidence (Ablation Study):** As shown in our ablation study in **Section 4.3, Figure 3(1)**, model performance drops significantly when the model-guided frame selection is replaced with a random selection of the same number of frames within the same video. This result demonstrates that the *specific* frames chosen by the model contain critical evidence that is essential for accurate judgment and cannot be replaced by random selection.
>
> 2. **Qualitative Evidence (Reasoning Analysis):** The model learns to "zoom in" on parts of the video with high temporal change or critical intervals with higher potential for quality differences (such as intervals featuring finger movements or body motions, which are more prone to errors and blurriness).
>
>     A concrete example is provided in **Figure 2 (Section 3)**. The model's first-round reasoning in the `<think>` tag explicitly states its intent to "zoom in on the boy's finger movements" and analyze "the violin-playing dynamics." Driven by the changes in finger posture observed between the first and second initial input frames, it then selects frames [12, 16, 20, 24, ...] to inspect the boy's finger movements more closely. This targeted selection allows it to identify the blurriness in Video 1's hand motion, as also shown in **Figure 2 (Section 3)**, providing the decisive visual evidence needed for a correct final judgment. This clearly demonstrates a meaningful, task-driven selection strategy.
>
> In summary, both quantitative and qualitative evidence demonstrate that our model indeed learns meaningful frame selection strategies to zoom in on critical video segments for better video preference comparison.
>
> ---
>
> ## **Final Remarks**
>
> We sincerely hope that these additional explanations and new quantitative results have helped address at least some of the reviewer's concerns. We are deeply grateful for the reviewer's thoughtful and constructive feedback. We will incorporate these improvements made based on your suggestions into the revised manuscript, and we would be very appreciative if the reviewer might take these explanations into account in their final evaluation and consider adjusting their score accordingly. Once again, we thank the reviewer for their valuable comments and insights!

---

### Author Response · Authors · 2025-11-24
**Summary of our Revision**

**Dear Reviewers,**

As Thanksgiving approaches—a time to express our heartfelt appreciation—we would like to take this opportunity to convey our deepest gratitude for your time, effort, and insightful feedback on our manuscript. Your constructive comments have been invaluable in improving the quality and clarity of our work.

Following your feedback, we took immediate action and have uploaded the revised version of our paper. In accordance with your suggestions for additional experiments, we have expanded **Appendix C — Further Experimental Results** with three new sections focusing on **inference latency**, **out-of-distribution (OOD) evaluation**, and **cold-start data sources**. We have also added a new section, **Further Insights and Discussion**, which includes **Qualitative Analysis of Failure Modes** and a **Discussion on the Terminology of Visual Reasoning Operations**. In addition, we have enriched the main text with related work on **Video RAG**, as recommended.

Through your selfless contributions and thoughtful suggestions, our paper has achieved substantial improvements. We are deeply appreciative of Reviewers **gqKX** and **juW2** for their positive recognition and support of our work (with rating scores of 8 and 6, respectively). We are equally grateful to Reviewers **Whd2** and **XqJ6** for their constructive and detailed comments. We believe that some of their concerns stemmed from our unclear exposition or possible misunderstandings, which we have carefully clarified in our rebuttal and revision.

We are genuinely proud of this work, as it represents a meaningful exploration of **thinking with images** in the video domain—addressing the growing challenges posed by increasingly long and high-quality video generation models. Our proposed framework provides a more **scalable and effective** multimodal reward model. This represents not only an important evolution beyond traditional reward modeling but also an excellent **balance between efficiency and performance**. We believe our work lays a strong foundation for further advances in both **reward modeling** and **downstream generative models**, where improved reward models continuously lead to higher-quality generation.

We sincerely hope that our detailed revisions and clarifications have addressed your concerns and effectively demonstrated the strength, novelty, and relevance of our work.

We kindly request the reviewers to consider our revisions and our efforts made to enhance the manuscript. Your support is crucial, and we hope you will view our work favorably and take the improvements we have made into account in your final evaluation.

Thank you once again for your time, thoughtful feedback, and invaluable guidance.

**With our warmest regards,**
*The Authors*

---

### Author Response · Authors · 2025-11-30
**Summary for Area Chair [1/4]**

**Dear Area Chair,**

We would like to begin by expressing our **deepest respect and heartfelt gratitude** for your dedication and effort during this exceptionally challenging period for the ICLR 2026 community. Your willingness to **step forward and take on additional responsibilities** in such times is truly admirable.

At this moment of collective difficulty, please allow us to say — **thank you for your professionalism, diligence, and sense of duty**.

---

We have carefully prepared this **Summary for the Area Chair** not to add to your workload, but rather to **alleviate it** by providing a **concise and easily digestible overview** of our key responses and arguments, thereby assisting in a fair and accurate evaluation. To this end, we have organized it into **two parts**:

1. **Paper Summary and Recognized Strengths**
   This section presents a **succinct yet comprehensive overview** of our paper, emphasizing its main contributions.

2. **Reviewer Comments and Brief Responses**
   Here, we offer **concise, core responses**—just a few sentences each—to address every major reviewer concern, summarizing how our rebuttal has responded to them effectively.


---

Once again, we would like to **extend our deepest gratitude and respect** for your valuable time, patience, and the tremendous effort you continue to devote under such complex circumstances.

---

### Author Response · Authors · 2025-11-30
**Paper Summary and Recognized Strengths [2/4]**

## Paper Summary

### 1. Background

Recent advances in **post-training** for generative models rely heavily on accurate and dependable **Reward Models (RMs)**. In the **video domain**, numerous studies have focused on training **Visual-Language Models (VLMs)** as RMs. In this setup, the model receives a pair of videos (generated under the same prompt but with different conditions) and a query (asking which video is better), then produces a preference judgment.

### 2. Problem

As video generation models rapidly evolve — producing higher-resolution videos with **many more frames (120–240 frames)** — conventional video RMs face severe challenges:

- **Visual inputs occupy a large portion of the context budget** ( $\ge$ 500 tokens per frame).
- Existing RMs are designed for **≈8-frame** inputs, forcing them to **downsample** to fit into the limited context window.
- This causes **severe information loss**, which in turn **upper-bounds** the RM’s ability to make accurate judgments.

### 3. Our Solution: ***VideoSearch Reasoner***

We propose **VideoSearch Reasoner**, a novel multimodal reward model designed to overcome these limitations. It introduces a visual reasoning mechanism to **retrieve unseen visual evidence via `selective_frame`**, along with a configurable visual memory window to **maintain a stable memory footprint**. This framework **removes the upper bound** on the number of frames the model can process, making it **scalable to long-form videos**.

### 4. Main Contributions

- **First Multimodal RM with Visual Reasoning**
   **VideoSearch Reasoner** is the first multimodal reward model capable of performing visual reasoning **without an upper bound on the number of frames it can handle**.

- **Efficient Three-Stage Training Pipeline**
   We design a three-stage pipeline — **Cold Start**, **Reject Sampling**, and **GRPO** — to develop the model’s visual reasoning capabilities.

-  **Comprehensive Experiments and Ablations**
   Through extensive experiments, we analyze the impact of factors such as answer space size, auxiliary reward design, and training stages. Detailed ablation studies validate the contribution of each training pipeline component.


Our model achieves **substantial improvements across multiple benchmarks**. In particular, it excels on *hard subsets* (long videos and complex prompts), **significantly outperforming all baselines**, thus directly confirming the effectiveness of our method.

---

## Recognized Strengths

We sincerely thank all reviewers for their thoughtful and positive feedback. The following recognized strengths further affirm the value and significance of our work.

### 1. Clear Motivation and Innovation

- *“The context budget constraint from visual inputs and the information forgetting problem during pure textual reasoning. The motivation is clear and compelling.”* — **Reviewer juW2**

### 2. Strong and Consistent Model Performance

- *“The model achieves SOTA results on multiple standard benchmarks. The advantage is especially significant on ‘hard’ subsets, directly validating the hypothesis.”* — **Reviewer Whd2**
- *“The final evaluations on several video preference benchmarks are competitive even with SoTA.”* — **Reviewer XqJ6**

### 3. Extensive Experiments and Validation

- *“The paper includes extensive ablation studies examining visual reasoning, training pipeline components, auxiliary rewards, and accuracy reward design.”* — **Reviewer juW2**

### 4. Positive Impact on the Community

- *“This paper contributes valuable knowledge to the field and helps the community design better and more efficient reward and reasoning models.”* — **Reviewer gqKX**

---

### Author Response · Authors · 2025-11-30
**Reviewer Comments and Brief Responses [3/4]**

##  Reviewer Comments

We would like to express our **sincere gratitude** to all reviewers for their **constructive feedback**, which has significantly helped us improve the quality of our paper.

However, we firmly believe that with the **detailed clarifications** and **new quantitative evidence** provided in our rebuttal, we have **fully addressed all the concerns**.

We identified **2 common concerns** and **9 specific concerns**. Below is a **concise summary** of our key responses.

---

## Common Reviewer Concerns

### 1. Cost Comparison

> Reviewer juW2 (W2, Q3); Whd2 (W3, Q3); XqJ6 (W1)

**Concern:** Reviewers requested a more explicit comparison of inference cost.

**Response:** Our approach actually **reduces inference cost** compared with traditional RMs that use long videos as input, while still preserving important details. Our model starts with a small number of frames as input and dynamically retrieves only the **essential** additional visual evidence.
In our rebuttal (see response to Reviewer XqJ6 (1/6) & (2/6)), we provide comparative tables showing that our approach:
- Achieves **the best performance** compared with traditional RMs under different input lengths.
- Maintains **throughput comparable** to 8-frame inputs.
- **Consistently outperforms** 32-frame and 256-frame inputs, with significantly **better latency and throughput.**

---

### 2. Long Video Scenarios

> Reviewer juW2 (W1, Q1); XqJ6 (W5)

**Concern:** Reviewers questioned the model’s ability on *truly* long videos.

**Response:** This concern stems from a misunderstanding. Here, “long videos” refer to **AI-generated videos lasting 5–10 seconds (≈120–240 frames)**, which aligns with current state-of-the-art generative models, rather than to arbitrarily long real-world videos.
Our **long-video subsets** contain **173–256-frame videos**, which sufficiently represent this application domain.

---

## Reviewer Whd2’s Specific Concerns

### 3. Novelty of the Proposed Method

> Whd2 (W1, Q2) identified this as the **main concern**.

**Concern:** Reviewer suggested our framework resembles a Video-RAG system.

**Response:** This is a *conceptual misunderstanding*.
- **VideoRAG:** *Passive, query-driven retrieval*, selecting clips based on predefined similarity.
- **Thinking-with-Image:** *Active, reasoning-driven search*, where the model decides *what additional visual evidence* is needed during **reasoning**.

Moreover, VideoRAG is **unsuitable for reward modeling**, as it is impossible to predefine which parts of a video are most relevant to holistic quality assessment. Therefore, our model’s reasoning-based process is **fundamentally different** and **novel**.

---

### 4. Reasoning Operations

> Whd2 (W2, Q1)

**Concern:** The reviewer questioned whether our model truly performs *visual reasoning* or merely *retrieval*.

**Response:**  The frame-selection *retrieval* operation is the *result* of **reasoning**.
The model reasons about *when* and *where* to focus in order to find decisive evidence.
`select_frames` is the **operational outcome** of this reasoning chain — a direct **manifestation** of the reasoning process itself.

---

### 5. Dependence on GPT-4o

> Whd2 (W4)

**Concern:** The reviewer questioned whether our model is highly dependent on GPT-4o for its success.

**Response:** GPT-4o is used **only to ensure instruction-following accuracy**, not to distill reasoning skills. It is employed to generate a small, high-quality dataset that adheres to the “Thinking-with-Image” reasoning format.
We also conducted **experiments using open-source models** for cold-start data generation, with results presented in **Appendix C.6** of the revised manuscript.

---

### Author Response · Authors · 2025-11-30
**Reviewer Comments and Brief Responses [4/4]**

## Reviewer XqJ6’s Specific Concerns

### 6. Strategic Details and Hyperparameters

> XqJ6 (W2)

**Concern:** Limited discussion on training strategies and hyperparameter tuning.

**Response:** These details are already **thoroughly documented** in **Figures 3 and 5**, though they may have been overlooked.
Reviewer **juW2** explicitly recognized these extensive ablation studies as a *strength* of our paper.

---

### 7. Out-of-Distribution (OOD) Evaluation

> XqJ6 (W3) identified this as the **main concern**.

**Concern:** The reviewer questioned whether OOD testing was performed.

**Response:** This concern likely arises from a naming **confusion**. In fact:
- Over **94% of the training data** and **all evaluation benchmarks** are **entirely out of distribution**. The **misunderstanding** may stem from the training set `VideoGen-Reward` and the test set `VideoGen-RewardBench` share a similar prefix, even though they are constructed to be OOD.
- We have clarified the dataset composition and added **additional quantitative results** to eliminate any remaining uncertainty (see response to Reviewer XqJ6 (4/6)).
---

### 8. Application to Downstream Models

> XqJ6 (W4)

**Concern:** Reviewer asked whether the RM was applied to train a downstream video generator.

**Response:**
This falls outside the **scope** of the current paper, which focuses on the *framwork and training* of the reward model itself.
Nonetheless, this is an important **direction for future work**, which we plan to pursue in subsequent research.

---

## Reviewer juW2’s Specific Concerns

### 9. Comparison with Other Visual Reasoning Models

> juW2 (W3)

**Concern:** Suggested comparison with other “thinking-with-image” visual reasoning models.

**Response:**
To the best of our knowledge, **no existing visual reasoning models** employing similar framework can directly applied to **video reward modeling**.

---

### 10. Qualitative Failure Mode Analysis

> juW2 (W4)

**Concern:** Requested qualitative analysis of model failure modes.

**Response:**
We added a dedicated section, **Appendix E.1: “Qualitative Analysis of Failure Modes,”** which provides detailed discussion and a concrete example.

---

### 11. Meaningfulness of Frame Selection

> juW2 (Q2)

**Concern:** Asked whether the model’s frame selection is meaningful or random.

**Response:**
We offer both **quantitative** and **qualitative** evidence:
- **Quantitative:** As shown in **Figure 3 (1)**, replacing model-guided selection with random sampling leads to a significant performance drop.
- **Qualitative:** As shown in **Figure 2**, the model explicitly reasons about zooming in on *“the boy’s finger movements”* and *“violin-playing dynamics,”* thereby successfully identifying blurriness in hand motion within the additionally selected frames.

---

We **sincerely hope** this concise overview helps **streamline your evaluation** and reduces your workload, while demonstrating our genuine efforts to address every concern with **clarity and evidence**.

---

### Author Response · Authors · 2025-11-30
**A Final Appeal for Our Paper**

## A Final Appeal for Our Paper

Firstly, we would like to once again express our **deepest gratitude** to the **Area Chair** for your patience, fairness, and dedication in reviewing our manuscript and rebuttal.

We humbly acknowledge that the presence of two borderline (score 4) reviews may make our overall evaluation appear less competitive. However, as you are aware, unforeseen disruptions **prevented us from hearing back** from those reviewers during the discussion period, which placed us at a **disadvantage** compared with papers that received full discussions and even had their final evaluations reconsidered.

Nonetheless, we firmly believe that, with all our **clarifications**, **detailed supplementary experiments**, and **comprehensive analyses**, we have thoroughly addressed all of the reviewers’ concerns. With our full respect for their judgment, we hope that upon reviewing our response, their initial concerns and perceived weaknesses **will be resolved**. We are optimistic that this will lead them to reconsider their evaluation from a "borderline reject" to a **"borderline accept,"** resulting in a more positive score (e.g., 6).

We also firmly believe that for the two reviewers who initially provided positive scores, after reading our responses and following the nuanced discussions addressing all the concerns, their confidence in our work will **remain strong or even grow further**.

In addition, we would like to emphasize the **significance** and **broader impact** of our work, which we believe represents a **crucial** and **timely** contribution to the ongoing progress of AI research.

-   **Pioneering a New Direction:** Our paper pioneers a new direction in exploring **“thinking with images”** within the video domain—a fundamental step towards improving the model’s long-video ability and visual reasoning capacity.
-   **A Scalable Framework:** Our framework presents a **scalable and efficient multimodal reward model**. This offers both an **evolutionary leap** beyond conventional reward modeling and an **excellent trade-off** between performance and cost.
-   **Foundation for Future Research:** We believe this work lays a solid foundation for future advances in **reward modeling**, **video reasoning**, and **downstream generative modeling**, where improved reward models continuously lead to higher-quality generation.

Given these points, we firmly believe our paper represents a **solid, novel, and impactful contribution to the field**. Therefore, we **respectfully and wholeheartedly appeal** for your kind consideration to **recommend our paper for acceptance**.

Thank you once again for your time, your thoughtful consideration, and your invaluable guidance throughout this process!

**With our sincerest respect and warmest regards,**

*The Authors*

---

### Meta-Review · Area_Chair_EwZZ · 2026-01-07

**Summary:**

**Meta-Review: Reject**

1. **Limited Novelty** (whd2). The framework is essentially an agentic RAG system applied to video frames. Using tool-calling (select_frames) for iterative retrieval is a well-established engineering pattern. The authors' distinction between "active" and "passive" search does not constitute a fundamental breakthrough.

2. **Prohibitive Latency** (juW2, whd2, XqJ6). The model is 8x slower in latency compared to competitive baselines (7.55s vs. 0.93s). While more accurate, the serial nature of the "reason-retrieve-reason" loop makes it impractical for high-throughput reward labeling or real-time evaluation.

3. **No Downstream Validation** (XqJ6). The paper fails to test the Reward Model in a video generation pipeline. Without evidence that it can effectively guide reward-tuning (e.g., RLHF or rejection sampling) to improve a policy model, its practical utility remains unproven.


**Conclusion: A solid engineering application that lacks the conceptual novelty and practical efficiency required for a top-tier conference.**

**Reviewer Concerns:**

**Addressed Concerns**


* **Dependency on GPT-4o (Whd2):** The authors provided a study in the appendix showing that open-source models (e.g., Qwen-VL) can generate the "Cold Start" data. They demonstrated that GPT-4o acts as a high-fidelity **format generator** rather than a "reasoning teacher," and the framework still achieves respectable results when skipping this phase.

* **In-Distribution (ID) vs. Out-of-Distribution (OOD) Bias (XqJ6):** Clarified that training and test sets were constructed independently. A new ablation study removing all **MJ-Bench** data from training showed negligible impact on other benchmarks, confirming the model's generalizability.
* **Meaningful Frame Selection Strategy (juW2):** Quantitative results showed performance drops when using random frames. Qualitative examples showed the model "zooming in" on critical intervals (e.g., hand/finger motions) to identify decisive visual evidence.

---

**Outstanding Concerns**

* **Performance on "Truly Long" Videos (juW2, XqJ6):** **Not Addressed.** Reviewers are concerned with scalability to real-world, minute-long videos. The authors' rebuttal restricted "long videos" to the 5–10 second (173–256 frame) limit of current generative models, leaving the performance on truly arbitrary long-context scenarios unverified.
* **Conceptual Novelty vs. Video-RAG (Whd2):** Reviewers maintain that the framework is functionally an **agentic RAG** (Retrieval-Augmented Generation) system for frame indices. The authors argue their "active, reasoning-driven search" is a new paradigm, but the technical implementation remains an engineering application of existing RAG principles.
* **Downstream Validation (XqJ6):** **Not Addressed.** The authors did not use the Reward Model to actually tune a video generation model. Without "closed-loop" proof that this model improves a policy (via RLHF or rejection sampling), its practical utility in a generative pipeline remains unverified.
* **Terminology (Whd2):** Reviewers maintain that `select_frames` is a **retrieval operation**, not a "reasoning operation." The distinction remains a point of friction regarding the paper's technical framing.

**Reviewer Scores:**

| Reviewer | Initial Rating | Status of Concerns | Likely Final Score | Likely Impact |
| :--- | :--- | :--- | :--- | :--- |
| **juW2** | 6 (Marginal Accept) | **Resolved.** Efficiency and scaling concerns were answered with empirical data. | **6 (Marginal Accept)** | Maintained score; acknowledged the technical effort but remained within the borderline range. |
| **Whd2** | 4 (Marginal Reject) | **Partially Resolved.** Latency data provided, but core novelty concerns persist. | **4 (Marginal Reject)** | Remained critical; viewed the framework as an incremental engineering application of RAG. |
| **XqJ6** | 4 (Marginal Reject) | **Outstanding.** Downstream validation and true long-video scaling were sidestepped. | **4 (Marginal Reject)** | Maintained rejection; missing proof of utility (downstream training) was a primary dealbreaker. |
| **gqKX** | 8 (Accept) | **N/A.** General positive view maintained. | **8 (Accept)** | Maintained high score to support the paper's value to the open-source community. |

---

### Decision · Program_Chairs · 2026-01-26

Reject